# On Learnability And Experience Replay Methods for Graph Incremental Learning on Evolving Graphs

## Abstract

Recent research has witnessed a surge in the exploration of Node-wise Graph Incremental Learning (NGIL) due to its substantial practical relevance. A central challenge in NGIL stems from the structural shifts induced by the inherent interdependence among vertices within graph data, adding complexity to the task of maintaining consistent model performance over time. Although several efforts have been made to devise incremental learning methods for NGIL, they have overlooked a fundamental question concerning the learnability of NGIL—whether there always exists a learning algorithm capable of consistently producing a model with a small error from the hypothesis. In this paper, we present the first theoretical study on the learnability of the NGIL problem with the statistical learning framework. Our analysis uncovers a critical insight: **NGIL is not always learnable when structural shifts are uncontrolled.** Additionally, in order to control structural shift, we leverage the idea of experience reply which selects a small set of representative data to replay with the new tasks, and propose a novel experience replay method, Structure-Evolution-Aware Experience Replay (SEA-ER). SEA-ER comprises a novel experience sample selection strategy founded on the topological awareness of GNNs and a novel replay objective utilizing importance re-weighting, which can effectively counteract catastrophic forgetting and mitigate the effect of structural shifts in NGIL. Comprehensive experiments validate our theoretical results and showcase the effectiveness of our newly proposed experience replay approach. Implementation is available at https://anonymous.4open.science/r/SEA-ER-8088/

## 1 Introduction

Graph neural networks (GNNs) have gained widespread popularity as effective tools for modelling graph and relational data structures (Wu et al., 2020). However, most studies on GNNs have predominantly focused on static settings, assuming that the underlying graph remains unchanged. This approach, while valuable for certain applications, does not reflect the dynamic nature of real-life networks, such as citation networks (Zliobaite, 2010) and financial networks (Gama et al., 2014), where graphs naturally evolve over time, accompanied by the emergence of new tasks. In practice, naively updating models with the latest data or tasks leads to a formidable issue known as *catastrophic forgetting* (Delange et al., 2021), wherein previously learned information is lost when learning the new tasks. Conversely, retraining models with the complete dataset each time new information arrives incurs excessive computational costs and often proves unfeasible. Thus, there is a pressing need to investigate graph incremental learning (GIL) (Febrinanto et al., 2023b), also known as graph continual learning or life-long graph learning, a field dedicated to developing learning algorithms that enable GNNs to continuously acquire new knowledge while retaining previously learned information.

Due to its practical significance, recent research has seen an upsurge in GIL investigations (Febrinanto et al., 2023a; Yuan et al., 2023). Nonetheless, there is a noticeable gap in the theoretical foundations, particularly regarding the node-level prediction task, commonly referred to as bf Node-wise Graph Incremental Learning (NGIL) (Su et al., 2023; Zhang et al., 2022). In the general formulation of NGIL, tasks arrive sequentially, and the GNN model undergoes multiple training rounds to

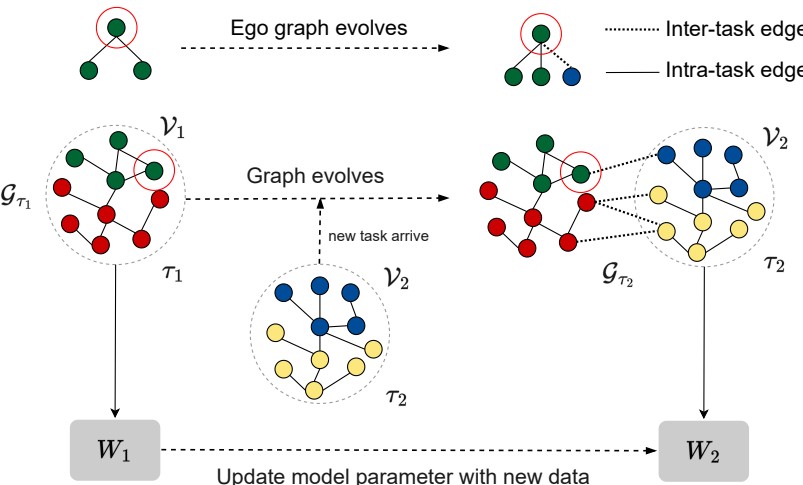

Figure 1: Illustration of the progression of NGIL. Task 2 introduces a new batch of vertices and results in an update to the parameters of the model from $W_2$ to $W_1$ using data from the new task. As new vertex batches associated with the new task $\tau_2$ are introduced, the graph structure changes, potentially altering the graph structure (inputs to GNNs) of the existing vertices, as captured by the changes in their ego graphs. This is referred to as the structural shift.

accumulate knowledge for specific objectives, such as node classification. In this scenario, vertices across different tasks can be interconnected, and the introduction of new tasks may alter the structure of the existing graph. Given that GNNs rely on graph structures as input, these structural changes can lead to shifts in the data distributions of previous tasks, a phenomenon known as *structural shift* (Su et al., 2023), as illustrated in Figure 1. *Before advancing methods for NGIL, it becomes imperative to determine the consistent learnability of the NGIL problem under structural shift.* Addressing this question regarding NGIL helps us understand the constraints, make informed decisions, and develop effective learning strategies. It is a critical facet of the NGIL problem that allows us to effectively address the challenges associated with evolving graph data.

In this work, we address the aforementioned question by formally formulating and analyzing the learnability of the NGIL problem. Our first contribution is highlighted as follows:

- *To the best of our knowledge, this work stands among the first theoretical studies on the learnability of NGIL.* We introduce a statistical learning perspective on NGIL, aiming to quantify how structural shifts impact the learnability of NGIL. Formally, we prove that without controls on the structural shift, NGIL is not always learnable, as articulated in Theorem 2.3. This means that there is no incremental learning algorithm that can consistently produce a good model (small error) from the hypothesis space, even if one exists in the hypothesis space. These insights suggest that incremental learning may not always be suitable for all dynamic graph systems.

Given the pivotal role of structural shifts in learnability, we recognize the importance of controlling structural shifts even in scenarios where incremental learning is feasible. To address this challenge, we leverage the idea of experience reply which has been empirically proven to be effective in addressing catastrophic forgetting in NGIL. Consequently, we introduce a novel experience replay framework designed to mitigate the impact of structural shifts in NGIL. Our contributions in this regard are as follows:

- We devise an innovative experience replay framework named Structure-Evolution-Aware Experience Replay (SEA-ER). SEA-ER addresses both catastrophic forgetting and the distinct challenge (structural shift) posed by NGIL, incorporating two key principles: 1) importance re-weighting and 2) topological awareness of GNNs. SEA-ER, tailored for GNNs, effectively mitigates structural shifts and catastrophic forgetting, offering theoretical assurances on long-term performance, as reflected in Proposition 3.1—a critical aspect of the incremental learning paradigm.

- We conduct an empirical evaluation encompassing real-world and synthetic datasets. The results align with our theoretical findings and demonstrate the effectiveness of our proposed approach. Our novel experience replay framework adeptly addresses both the structural shift and catastrophic forgetting challenges, outperforming contemporary GNN benchmarks (Zhou & Cao, 2021a; Ahrabian et al., 2021; Kim et al., 2022) in NGIL.

## 2 LEARNABILITY OF NGIL

### 2.1 PRELIMINARY

**Notations.** We use bold letters to denote random variables, while the corresponding realizations are represented with thin letters. NGIL assumes the existence of a stream of training tasks $\tau_1, \tau_2, ..., \tau_m$, characterized by observed vertex batches $\mathcal{V}_1, \mathcal{V}_2, ..., \mathcal{V}_m$. Each vertex $v$ is associated with a node feature $x_v \in \mathcal{X}$ and a target label $y_v \in \mathcal{Y}$. The observed graph structure at training task $\tau_i$ is induced by the accumulative vertices and given by $\mathcal{G}_{\tau_i} = \mathcal{G}[\bigcup_{j=1}^{i} \mathcal{V}_j]$. In this setting, the graph structure is evolving as the learning progresses through different training tasks, as illustrated in Fig. 1.

Node-level learning tasks with GNNs rely on information aggregation within the k-hop neighbourhood of a node as input. To accommodate for this nature, we adopt a local view in the learning problem formulation as in (Su et al., 2023). We denote $\mathcal{N}_k(v)$ as the k-hop neighborhood of vertex $v$, and the nodes in $\mathcal{N}_k(v)$ form an ego-graph $g_v$, which consists of a (local) node feature matrix $X_v = \{x_u | u \in \mathcal{N}_k(v)\}$ and a (local) adjacency matrix $A_v = \{a_{uw} | u, w \in \mathcal{N}_k(v)\}$. We denote $\mathbb{G}$ as the space of possible ego-graph and $\mathbf{g}_v$ as a random variable of the ego-graph for the target vertex $v$, whose realization is $g_v = (A_v, X_v)$. Let $g_\mathcal{V} = \{g_v | v \in \mathcal{V}\}$ denote the set of ego graphs associated with vertex set $\mathcal{V}$. From the perspective of GNNs, the ego-graph $g_v$ is the Markov blanket containing all necessary information for the prediction problem for the root vertex $v$. Therefore, we can see the prediction problem associated with data $\{(g_v, y_v)\}_{v \in \mathcal{V}_i}$ from training session $\tau_i$ as drawn from a joint distribution $\mathbb{P}(\mathbf{y_v}, \mathbf{g}_v | \mathcal{V}_i)$.

Let $\mathcal{F}$ denote the hypothesis space and $f \in \mathcal{F}$ be a classifier with $\hat{y}_v = f(g_v)$ and $\mathcal{L}(.,.) \mapsto \mathbb{R}$ be a given loss function. We use $R^{\mathcal{L}}_{\mathbb{P}(\mathbf{y}_v, \mathbf{g}_v | \mathcal{V})}(f)$ to denote the generalization risk of the classifier $f$ with respect to $\mathcal{L}$ and $\mathbb{P}(\mathbf{y}_v, \mathbf{g}_v | \mathcal{V})$, and it is defined as follows:

$$R^{\mathcal{L}}_{\mathbb{P}(\mathbf{y}_v, \mathbf{g}_v | \mathcal{V})}(f) = \mathbb{E}_{\mathbb{P}(\mathbf{y}_v, \mathbf{g}_v | \mathcal{V})}[\mathcal{L}(f(g_v), y_v)]. \tag{1}$$

**Catastrophic Forgetting.** With the formulation above, the catastrophic forgetting (**CF**) of a classifier $f$ after being trained on $\tau_m$ can be characterized by the retention of performance on previous vertices, given by:

$$\mathbf{CF}(f) := R^{\mathcal{L}}_{\mathbb{P}(\mathbf{y}_v, \mathbf{g}_v | \mathcal{V}_1, ..., \mathcal{V}_{m-1})}(f), \tag{2}$$

which translates to the retention of performance of the classifier $f$ from $\tau_m$ on the previous tasks $\tau_1, ..., \tau_{m-1}$.

### 2.2 IMPOSSIBILITY RESULT OF NGIL

In this paper, we focus on the effect of structural shifts in the learnability of NGIL. Next, we show the main theoretical result of this paper, which states that having no control over the change in graph structure renders the NGIL unlearnable, despite the hypothesis space (as captured by GNNs) being powerful. We begin with defining the expressiveness of hypothesis space and characterizing the structural shift.

**Definition 2.1** (expressiveness). For a set of distributions $\{\mathbb{P}_1, \mathbb{P}_2, ..., \mathbb{P}_k\}$ and a hypothesis space $\mathcal{F}$, we define the joint prediction error of the hypothesis space on the given distributions as,

$$\mathcal{E}_\mathcal{F}(\{\mathbb{P}_1, \mathbb{P}_2, ..., \mathbb{P}_k\}) = \inf_{f \in \mathcal{F}} [\sum_{\mathbb{P} \in \{\mathbb{P}_1, \mathbb{P}_2, ..., \mathbb{P}_k\}} R^{\mathcal{L}}_{\mathbb{P}}(f)].$$

The joint prediction error characterizes the capability (expressiveness) of the hypothesis space to simultaneously fit a given set of distributions. Obviously, the expressiveness of the given hypothesis

space would be a trivial necessary condition related to how learnable a given setting is. If the expressiveness of the hypothesis is not powerful enough, the best classifier in the hypothesis space would still suffer from a bad performance. For the remaining paper, we focus our analysis on the structural shift and assume the expressive power of the hypothesis space is powerful enough, i.e., there exists a classifier in the hypothesis that incures a small error. We provide a further discussion on the expressiveness and sample complexity in Appendix A.

Next, we formalize the notion of a structural shift among different vertex batches.

**Definition 2.2.** Let $\mathbb{P}_1$ and $\mathbb{P}_2$ be two distributions over some domain $\mathcal{X}$, $\mathcal{F}\Delta\mathcal{F}$-distance between $\mathbb{P}_1$ and $\mathbb{P}_2$ is defined as,

$$d_{\mathcal{F}\Delta\mathcal{F}}(\mathbb{P}_1, \mathbb{P}_2) = 2 \sup_{A \in \mathcal{F}\Delta\mathcal{F}} \|\mathbb{P}_1(A) - \mathbb{P}_2(A)\|,$$

where $\mathcal{F}\Delta\mathcal{F} = \{\{x \in \mathcal{X} : f(x) \neq f'(x)\} : f, f' \in \mathcal{F}\}$.

$\mathcal{F}\Delta\mathcal{F}$-distance is commonly used in domain adaptation to capture the relatedness (distance) of distributions with respect to the hypothesis space $\mathcal{F}$ (Redko et al., 2020). Here, we adopt $\mathcal{F}\Delta\mathcal{F}$-distance to characterize the effect of the emerging task on the distribution of the previous vertex tasks. If the structural shift among different tasks is large, then the appearance of the new vertex batch would shift the distribution of the previous vertex batch to a larger extent. As a result, the $\mathcal{F}\Delta\mathcal{F}$-distance between the updated distribution and the previous distribution would be large.

For the analysis, we focus on the effect of structural shift on catastrophic forgetting and assume that the labelling rule stays the same for vertices in different sessions, i.e., $\mathbb{P}(\mathbf{y}|\mathbf{g}_v, \tau_i) = \mathbb{P}(\mathbf{y}|\mathbf{g}_v, \tau_j)$, $\forall i, j$. Without loss of generality, we may gauge our learnability analysis toward the case of two training sessions, referred to as the NGIL-2 problem. If the NGIL-2 problem is not learnable, then trivially, the general NGIL-k is not learnable. The NGIL-2 problem is characterized by three distributions: $\mathbb{P}(\mathbf{y}, \mathbf{g}_v|\mathcal{V}_1, \mathcal{G}_{\tau_1})$, $\mathbb{P}(\mathbf{y}, \mathbf{g}_v|\mathcal{V}_1, \mathcal{G}_{\tau_2})$, and $\mathbb{P}(\mathbf{y}, \mathbf{g}_v|\mathcal{V}_2, \mathcal{G}_{\tau_2})$, which are the distributions of $\mathcal{V}_1$ in graphs $\mathcal{G}_{\tau_1}$ and $\mathcal{G}_{\tau_2}$, and the distribution of $\mathcal{V}_2$ in graph $\mathcal{G}_{\tau_2}$.

**Theorem 2.3** (learnability of NGIL, informal). *Let $\mathcal{F}$ be a hypothesis space over $\mathbb{G} \times \mathcal{Y}$. For every $c > 0$, if there is no control of $d_{\mathcal{F}\Delta\mathcal{F}}(\mathbb{P}(\mathbf{y}, \mathbf{g}_v|\mathcal{V}_1, \mathcal{G}_{\tau_1}), \mathbb{P}(\mathbf{y}, \mathbf{g}_v|\mathcal{V}_2, \mathcal{G}_{\tau_2}))$ or $d_{\mathcal{F}\Delta\mathcal{F}}(\mathbb{P}(\mathbf{y}, \mathbf{g}_v|\mathcal{V}_1, \mathcal{G}_{\tau_2}), \mathbb{P}(\mathbf{y}, \mathbf{g}_v|\mathcal{V}_2, \mathcal{G}_{\tau_2}))$, then there exists an instance of NGIL-2 problem such that even if $\mathcal{E}_{\mathcal{F}}(\{\mathbb{P}(\mathbf{y}, \mathbf{g}_v|\mathcal{V}_1, \mathcal{G}_{\tau_1}), \mathbb{P}(\mathbf{y}, \mathbf{g}_v|\mathcal{V}_1, \mathcal{G}_{\tau_2})\}) \leq c$, NO learning algorithm that can produces a good classifier $f$ from $\mathcal{F}$ with probability greater than or equal to $1/2$.*

The formal version and proof of Theorem 2.3 can be found in Appendix A. The main idea of the proof is to identify and establish the intrinsic connection between the NGIL problem and the domain adaptation problem. Theorem 2.3 states that even if the hypothesis space is expressive enough (i.e., there exist classifiers incurring an arbitrary small joint prediction error), the NGIL problem might still not be learnable if there is no control on the structural shift among different vertex batches. Theorem 2.3 offers important insights for making well-informed decisions. To illustrate, Theorem 2.3 provides a clear indication that when the underlying graph structure undergoes rapid and frequent changes within a network system, it may not be advisable to pursue incremental learning for updating the machine learning model associated with that system. Theorem 2.3 also signifies the pivotal role of the structural shifts in the learnability of NGIL. This implies the importance of controlling structural for NGIL even in the learnable scenario and motivates our proposed method in the next section.

## 3 STRUCTURE-EVOLUTION-AWARE EXPERIENCE REPLAY (SEA-ER)

In this section, we present SEA-ER which combines the topology awareness of GNNs and the importance re-weighting for addressing the catastrophic forgetting and structural shift in the NGIL problem. The idea of experience replay is to select a small subset of representative data samples from the past and to replay them in each new learning task to consolidate past knowledge. Experience replay methods involve two main components, 1) experience buffer selection: selecting samples from the past to be stored in the experience buffer, and 2) replay strategy: deciding how the experience buffer is replayed (trained) with the new task. The pseudo-code of our methods can be found in Appendix F.

## 3.1 Topoligical-Aware Experience Buffer Selection

We denote the experience buffer for task $\tau_i$ as $\mathcal{P}_i = \{P_1, P_2, ..., P_{i-1}\}$, where $P_j$ represents samples selected from task $\tau_j$. GNNs have been shown to exhibit better generalization performance on vertices with closed structural similarity (Ma et al., 2021). This means that a vertex set with close structural similarity to the rest of the vertices is more representative of the graph. Based on this idea, we propose an experience buffer selection strategy that selects samples with the closest structural similarity to the remaining vertices. This strategy amounts to the following optimization problem:

$$\underset{P_j \subset \mathcal{V}_j}{\arg\min} \max_{u \in \mathcal{V}_j \setminus P_j} \min_{v \in P_j} d_{\text{spd}}(u, v), \quad s.t. \quad |P_j| = b. \tag{3}$$

Here, $d_{\text{spd}}(.)$ represents the shortest path distance, and $b$ is the number of samples to select from each task. The optimization above returns a set $P_j$ with close structural similarity to the rest of the vertices within the same task. Eq. 3 amounts to the k-center problem in the graph and is NP-hard. While finding the optimal solution to this problem may be intractable, efficient heuristics can be employed to tackle it. For our experiments, we adopt the classic farthest-first traversal algorithm to obtain a solution.

## 3.2 Replay Strategy with Importance Reweighting

The standard replay strategy includes the experience buffer in the training set of the newest task, treating the samples in the experience buffer as normal training samples. Suppose we are at task $\tau_i$ with training set $\mathcal{V}_i^{\text{train}}$, the standard replay strategy amounts to the following learning objective,

$$\mathcal{L} = \frac{1}{|\mathcal{V}_i^{\text{train}}|} \sum_{v \in \mathcal{V}_i^{\text{train}}} \mathcal{L}(v) + \sum_{P_j \in \mathcal{P}_i} \frac{1}{|P_j|} \sum_{v \in P_j} \mathcal{L}(v), \tag{4}$$

where $\mathcal{L}(v)$ is the loss on vertex $v$. As a result, the standard replay strategy weights the samples in the new task and experience buffer equally.

Nevertheless, as we discussed earlier, the structural shifts induced by the evolving graph structure ($\mathcal{G}_{\tau_{i-1}} \rightarrow \mathcal{G}_{\tau_i}$) can alter the distribution of vertices from the previous task, which can lead to sub-optimal learning. To address the issue, we propose a novel replaying strategy with importance re-weighting (David et al., 2010). This strategy assigns higher weights to replay samples that are more similar to the distribution before the structural shift, ensuring that the model pays more attention to relevant information. The learning objective for replay with importance re-weighting is as follows:

$$\mathcal{L} = \frac{1}{|\mathcal{V}_i^{\text{train}}|} \sum_{v \in \mathcal{V}_i^{\text{train}}} \mathcal{L}(v) + \sum_{P_j \in \mathcal{P}_i} \frac{1}{|P_j|} \sum_{v \in P_j} \beta_v \mathcal{L}(v), \tag{5}$$

where $\beta_v$ is the weight for vertex $v$ in the experience buffer. To determine $\beta$ (collection of $\beta_v$), we use kernel mean matching (KMM) Gretton et al. (2009) with respect to the embeddings of vertices in the experience buffer. This involves solving the following convex quadratic problem:

$$\min_{\beta} \left\| \sum_{P_j \in \mathcal{P}_i} \sum_{v \in P_j} \beta_v \phi(h_v) - \phi(h'_v) \right\|^2, \quad s.t. \quad B_l \leq \beta_v < B_u, \tag{6}$$

where $h_v$ is the node embedding of $v$ (output of the GNN) with respect to $\mathcal{G}_{\tau_i}$ and $h'_v$ is the node embedding of $v$ with respect to $\mathcal{G}_{\tau_{i-1}}$. Eq. 6 is equivalent to match the mean elements in some kernel space $\mathcal{K}(.,.)$ and $\phi(.)$ is the feature map to the reproducing kernel Hilbert space induced by the kernel $\mathcal{K}(.,.)$. In our experiment, we use a mixture of Gaussian kernel $\mathcal{K}(x, y) = \sum_{\alpha_i} e^{-\alpha_i ||x-y||_2}$, $\alpha_i = 1, 0.1, 0.01$. The lower bound $B_l$ and upper bound $B_u$ constraints are to ensure reasonable weight for most of the instances.

## 3.3 Theoretical Analysis

In this section, we delve into the theoretical underpinnings of our proposed method by introducing the concept of distortion rate, which aids in characterizing the topological awareness of GNNs.

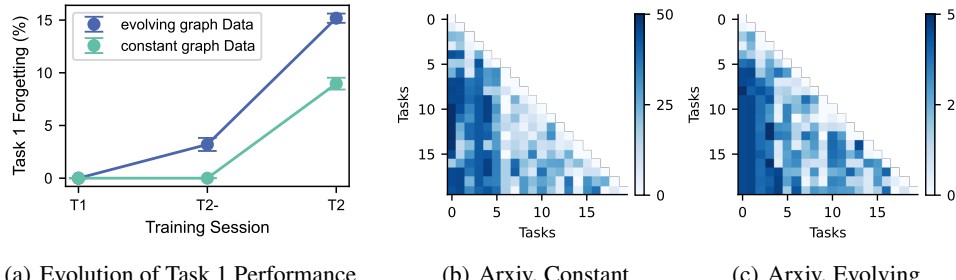

(a) Evolution of Task 1 Performance    (b) Arxiv, Constant    (c) Arxiv, Evolving

Figure 2: Forgetting Dynamics of Bare Model on Arxiv under Settings with Constant and Evolving Graphs. (a) captures the change of catastrophic forgetting of task 1 when transitioning into task 2 under the settings with constant and evolving graphs. (b) and (c) are the complete catastrophic forgetting matrix ($x, y$-axis are the $i, j$ in $r_{i,j} - r_{i,i}$ correspondingly) of Bare model in the inductive and transductive settings.

Table 1: Performance comparison of SEA-ER to existing experience replay NGIL frameworks (↑ higher means better). Results are averaged among five trials. **Bold letter** indicates that the entry admits the best performance for that given dataset. 5% of the training set size is used for the size of the experience replay buffer for all the experience replay methods.

| Dataset | CoraFull | | OGB-Arxiv | | Reddit | |
|---|---|---|---|---|---|---|
| Performance Metric | FAP (%) ↑ | FAF (%) ↑ | FAP (%) ↑ | FAF (%) ↑ | FAP(%) ↑ | FAF (%) ↑ |
| Bare model | $61.41 \pm 1.2$ | $-30.36 \pm 1.3$ | $59.24 \pm 3.6$ | $-33.72 \pm 3.3$ | $68.60 \pm 1.8$ | $-23.87 \pm 1.7$ |
| Joint Training | $92.34 \pm 0.6$ | N.A. | $94.36 \pm 0.4$ | N.A. | $95.27 \pm 1.2$ | N.A. |
| ER:IID-Random | $77.24 \pm 5.8$ | $-16.73 \pm 5.6$ | $79.72 \pm 4.6$ | $-14.89 \pm 4.5$ | $80.08 \pm 3.3$ | $-10.89 \pm 2.5$ |
| ER-deg | $86.24 \pm 0.7$ | $-8.73 \pm 0.7$ | $85.28 \pm 0.8$ | $-10.89 \pm 1.9$ | $85.08 \pm 0.9$ | $-8.89 \pm 0.9$ |
| ER-infl | $87.99 \pm 0.6$ | $-8.31 \pm 0.5$ | $86.42 \pm 1.2$ | $-7.65 \pm 1.3$ | $86.98 \pm 1.4$ | $-6.82 \pm 1.6$ |
| ER-rep | $88.95 \pm 0.9$ | $-7.55 \pm 0.8$ | $88.12 \pm 0.9$ | $-8.23 \pm 0.9$ | $86.02 \pm 1.2$ | $-7.21 \pm 1.3$ |
| SEA-ER w.o. ir (ours) | $89.13 \pm 0.8$ | $-7.21 \pm 0.9$ | $89.19 \pm 0.7$ | $-7.13 \pm 0.5$ | $88.48 \pm 1.7$ | $-6.10 \pm 1.6$ |
| ER:IID-Random w. ir | $80.24 \pm 6.9$ | $-14.73 \pm 4.8$ | $81.72 \pm 5.2$ | $-13.23 \pm 5.2$ | $83.43 \pm 2.8$ | $-10.32 \pm 3.8$ |
| ER-deg w. ir | $87.32 \pm 0.5$ | $-7.89 \pm 0.5$ | $86.88 \pm 0.6$ | $-9.32 \pm 2.0$ | $87.11 \pm 1.0$ | $-6.89 \pm 1.0$ |
| ER-infl w. ir | $88.88 \pm 0.5$ | $-8.01 \pm 0.5$ | $89.42 \pm 1.3$ | $-5.65 \pm 1.7$ | $90.64 \pm 1.6$ | $-3.63 \pm 1.3$ |
| ER-rep w. ir | $89.06 \pm 0.7$ | $-7.00 \pm 0.9$ | $88.92 \pm 0.8$ | $-8.00 \pm 1.1$ | $88.99 \pm 1.4$ | $-4.21 \pm 1.1$ |
| SEA-ER (ours) | $\mathbf{91.67} \pm 1.3$ | $\mathbf{-4.01} \pm 1.4$ | $\mathbf{92.88} \pm 1.1$ | $\mathbf{-3.08} \pm 0.7$ | $\mathbf{92.89} \pm 1.5$ | $\mathbf{-2.72} \pm 1.4$ |

A GNN is said to have a distortion rate $\alpha$ if there exists a constant $r > 0$ such that for all $u, v \in \mathcal{V}$, the following inequalities hold: $\forall u, v \in \mathcal{V}, rd_{\text{spd}}(u, v) \leq d(h_u, h_v) \leq \alpha rd_{\text{spd}}(u, v)$. The distortion rate quantifies the ratio of distances in the embedding space to those in the original graph space. A low distortion rate, close to 1, signifies that the embedding effectively preserves the structural information from the original graph. This measure allows us to assess the topological awareness of GNNs and forms the basis for our theoretical analysis of the experience selection strategy in relation to the underlying graph topology.

**Proposition 3.1.** *Given a collection task $\tau_1, ..., \tau_m$ and their associated vertex set $\mathcal{V}_1, ..., \mathcal{V}_m$. Let $\mathcal{P}_m = \{P_1, ..., P_{m-1}\}$ be the experience buffer where each $P_i \in \mathcal{P}_m$ has the same cardinality and cover all classes in $\tau_i$. Let $g$ be the GNN model with distortion rate $\alpha$. Assuming the loss function $\mathcal{L}$ is bounded, continuous and smooth, we have that,*

$$R^{\mathcal{L}}_{\mathbb{P}(\mathbf{y}_v, \mathbf{g}_v | \mathcal{V}_i \backslash P_i)}(f) \leq R^{\mathcal{L}}_{\mathbb{P}(\mathbf{y}_v, \mathbf{g}_v | P_i)}(f) + \mathcal{O}(\alpha D_s(\mathcal{V}_i \backslash P_i, P_i)),$$

*where $D_s(\mathcal{V}_i \backslash P_i, P_i) = \max_{u \in \mathcal{V}_i \backslash P_i} \min_{v \in P_i} d_{\text{spd}}(v, u)$.*

The proof for Propostion 3.1 can be found in Appendix C. This proposition illuminates that catastrophic forgetting in the model, as measured by its ability to retain performance on past data, is bounded by the training loss on the experience buffer and is further influenced by a combination of factors involving the distortion rate of the model and the structural distance within the experience buffer. This underscores the critical role of graph structure when selecting a suitable experience buffer, as the structural distance substantially impacts the performance bound. Consequently, Proposition 3.1 validates the optimization objective, Eq. 3, for selecting an experience buffer and provides a performance guarantee for our proposed experience selection strategy.

## 4 EXPERIMENTS

In this section, we present an empirical validation of our theoretical results and evaluation of our proposed experience replay method for NGIL. Through a series of experiments, we aim to answer the following questions: (1) Does the structural shifts have an impact on the learnability and the performance of GNNs in NGIL? (2) Does the distortion rate of GNNs, as discussed in Theorem 1, hold true in practice? (3) How does our proposed experience replay framework compare to existing experience replay methods for GNNs (Zhou & Cao, 2021a; Ahrabian et al., 2021; Kim et al., 2022)? Due to space limitations, we provide a more comprehensive description of the datasets, experiment set-up and additional results in Appendix D.

**Datasets and Experimental Set-up.**
We conduct experiments on both real-world and synthetic datasets. Real-world datasets include OGB-Arxiv (Hu et al., 2020), Reddit (Hamilton et al., 2017), and CoraFull (Bojchevski & Günnemann, 2017). To generate the synthetic dataset, we use the popular contextual stochastic block model (cSBM) (Deshpande et al., 2018)

| Datasets | OGB-Arxiv | Reddit | CoraFull |
|---|---|---|---|
| # vertices | 169,343 | 227,853 | 19,793 |
| # edges | 1,166,243 | 114,615,892 | 130,622 |
| # class | 40 | 40 | 70 |
| # tasks | 20 | 20 | 35 |
| # vertices / # task | 8,467 | 11,393 | 660 |
| # edges / # task | 58,312 | 5,730,794 | 4,354 |

Table 2: Incremental learning settings for each dataset.

(further details of the cSBM can be found in Appendix D.4). The experimental set-up follows the widely adopted task-incremental-learning (task-IL)(Zhang et al., 2022), where a k-class classification task is extracted from the dataset for each training session. For example, in the OGB-Arxiv dataset, which has 40 classes, we divide them into 20 tasks: Task 1 is a 2-class classification task between classes 0 and 1, task 2 is between classes 2 and 3, and so on. In each task, the system only has access to the graph induced by the vertices at the current and earlier learning stages, following the formulation in Sec.2.1. A brief description of the datasets and how they are divided into different node classification tasks is given in Table 2. We adopt the implementation from the recent NGIL benchmark (Zhang et al., 2022) for creating the task-IL setting and closely following their set-up (such as the train/valid/test set split).

**Evaluation Metrics.** Let $r_{i,j}$ denote the performance (accuracy) on task $j$ after the model has been trained over a sequence of tasks from 1 to $i$. Then, the forgetting of task $j$ after being trained over a sequence of tasks from 1 to $i$ is measured by $r_{i,j} - r_{j,j}$. We use the final average performance (FAP) := $\frac{\sum_j^m r_{m,j}}{m}$ and the final average forgetting (FAF) := $\frac{\sum_j^m r_{m,j} - r_{j,j}}{m}$ to measure the overall effectiveness of an NGIL framework.

**Baselines.** We compare the performance of SEA-ER with the following experience replay baselines. All the baselines are using the standard training objective given in Eq. 4. **ER-IID-Random** updates each task with an experience buffer with vertices randomly selected from each task. **ER-IID-Random** serves as a baseline for showing the effectiveness of different experience buffer selection strategies. We adopt state-of-the-art experience replay frameworks for GNNs for comparison. **ER-rep** (Zhou & Cao, 2021a) selects an experience buffer based on the representations of the vertex. **ER-deg** (Ahrabian et al., 2021) selects an experience buffer selected based on the degree of vertices. **ER-infl** (Kim et al., 2022) selects an experience buffer selected based on the influence score of the representations of the vertex. We adopt the implementation of the two baselines (Ahrabian et al., 2021; Zhou & Cao, 2021a) from the benchmark paper (Zhang et al., 2022) and implement a version of (Kazemi et al., 2020) based on our understanding, as the code is not open-sourced. **SEA-ER w.o. ir** is our proposed experience replay selection strategy with the standard learning objective Eq. 4, and **SEA-ER** is our proposed experience replay selection strategy with learning objective Eq. 5. The suffix **w. ir** indicate the usage of Eq. 5. In addition to the above frameworks, we also include two natural baselines for NGIL: the **Bare model** and **Joint Training**. The **Bare model** denotes the backbone GNN without any continual learning techniques, serving as the lower bound on continual learning performance. On the other hand, **Joint Training** trains the backbone GNN on all tasks simultaneously, resulting in no forgetting problems and serving as the upper bound for continual learning performance.

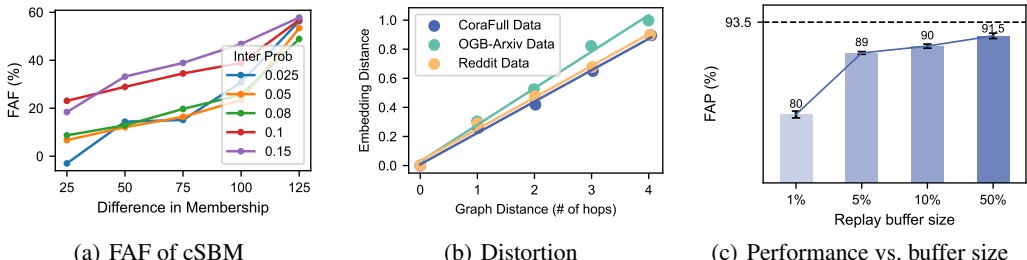

(a) FAF of cSBM  (b) Distortion  (c) Performance vs. buffer size

Figure 3: Fig. 3(a) is the experiments with different configurations of cSBM models. Fig. 3(b) is the distortion between graph distance and embedding distance. Fig. 3(c) is the experiment on the effect of replay buffer size.

## 4.1 RESULTS

**Impact of Structural Shift.** We first show the effect of evolving graph structure on the incremental learning system. We do so by comparing the dynamic of forgetting (change of $r_{i,j} - r_{i,i}$ for different $i, j$) of the Bare model. The result of the OGB-Arxiv dataset is illustrated in Fig. 2(a). In Fig. 2(a), $T_1$ and $T_2$ on the x-axis denote the first and the second tasks (training sessions), respectively, and $T_2-$ represents the state when Task 2 has arrived but the model has not been trained on Task 2 data yet. As shown in the figure, in the case of an evolving graph, the catastrophic forgetting on Task 1 occurs twice when entering stage 2: when vertices of Task 2 are added to the graph (structural shift) and when the model is trained on Task 2's data. Fig. 2(b) and Fig. 2(c) are the complete catastrophic forgetting matrix (with $r_{i,j} - r_{i,i}$ as entries) for the two settings. The colour density difference of the two performance matrices illustrates that the existing tasks suffer more catastrophic forgetting in the setting with evolving graphs. The results of this experiment align with previous findings, offering a complementary view from the perspective of forgetting.

We then further quantify the structural relation between the structural shift and the performance of an incremental learning system. We use cSBM (further details can be found in Appendix D.4) to create a two-session NGIL with a two-community graph of the size 300 for each training session, and the task is to differentiate and classify vertices between the two communities. The community membership configuration is described by a four-tuple $(c_0^{(1)}, c_1^{(1)}, c_0^{(2)}, c_1^{(2)}) = (150 + \delta, 150 - \delta, 150 - \delta, 150 + \delta)$, where $c_i^{(j)}$ is the number of vertices from community $i$ at session $j$. Then, we vary the structural shift by changing (1) the inter-connectivity $p_{\text{inter}}$ (probability of an edge between tasks/sessions) and (2) the difference ($\delta$) of each community member in different stages. Intuitively, as we increase $\delta$ and the inter-connectivity $p_{\text{inter}}$, the structural shift between the two training sessions increases and therefore we should observe a larger FAF converging to random guess as predicted by the theoretical results. As shown in Fig. 3(a), this is indeed the case.

**Distortion Rate.** We next validate that the distortion rate of GNNs is small. We train a vanilla GraphSAGE model using the default setting for each dataset. We extract vertices within the 5-hop neighbourhood of the training set and group the vertices based on their distances to the training set. We then compute the average embedding distance of each group to the training set in the embedding space. In Fig. 3(b), the distortion rate and scaling factor of the GNN are reflected by the slope of the near-linear relation between the embedding distance and the graph distance. We can see that the distortion factors in the performance bound in Proposition 3.1 are indeed small (closed to 1), validating our proposed experience selection strategy.

**Effectiveness of SEA-ER.** In Table 1, we show the FAP and FAF of each method after learning the entire task sequence. On average, the Bare model without any continual learning technique performs the worst, and Joint training performs the best. FAF is inapplicable to joint-trained models because they do not follow the continual learning setting and are simultaneously trained on all tasks. Our experience buffer selection strategy, SEA-ER w.o. ir, outperforms ER-rep, ER-infl and ER-deg in the proposed setting. In addition, the modified learning objective Eq. 5 with importance re-weighting can boost the performance of all experience replay frameworks by adjusting the structural shifts induced by the evolving graph structure.

**Ablation Study on the Size of Experience Replay Buffer.** As illustrated in Fig. 3(c), the performance of SEA-ER-re, predictably, converges to the Joint Training setting as the size of the experi-

ence replay buffer increases. Nonetheless, the size of the experience replay buffer is often subject to system constraints, such as the available storage capacity. The experiment demonstrates that SEA-ER is highly effective in preserving the performance of the system (as indicated by the final average task performance, FAP), even when the size of the experience replay buffer is limited.

## 5 RELATED WORK

Due to space constraints, we provide a brief overview of related work in the main paper and refer interested readers to Appendix E for a more comprehensive discussion on incremental learning and GIL in general.

Recent strides in GNNs have stimulated increased exploration in GIL, owing to its pragmatic relevance (Wang et al., 2022; Xu et al., 2020; Daruna et al., 2021; Kou et al., 2020; Ahrabian et al., 2021; Cai et al., 2022; Wang et al., 2020; Liu et al., 2021; Zhang et al., 2021; Zhou & Cao, 2021b; Carta et al., 2021; Kim et al., 2022; Su et al., 2023). For a detailed examination of GIL methodologies and their efficacy, we direct readers to recent reviews and benchmarks (Zhang et al., 2022; Yuan et al., 2023; Febrinanto et al., 2023a). However, it's noteworthy that most existing NGIL studies primarily focus on scenarios where the entire graph structure is predefined or where interconnections between tasks are disregarded. The NGIL setting, where the graph evolves with the introduction of new tasks, remains relatively unexplored and warrants a deeper theoretical investigation. To our knowledge, the sole existing theoretical study on NGIL with an evolving graph is presented in (Su et al., 2023), where they provide an upper bound of structural shift on catastrophic forgetting. In contrast, our research endeavors to further address the challenges of NGIL with an evolving graph by presenting the first impossibility result regarding the learnability of the NGIL problem, underscoring the critical role of structural shifts. This impossibility result can be interpreted as a "lower bound" on the extent of structural shift that can induce catastrophic forgetting, complementing the findings in (Su et al., 2023) and emphasizing the pivotal role of structural shifts in NGIL. Additionally, we introduce a novel experience replay method that effectively tackles both catastrophic forgetting and structural shifts in NGIL.

Finally, it should be noted that there is another orthogonal line of research known as dynamic or temporal graph learning, which focuses on developing GNNs capable of capturing changing graph structures. A comprehensive review of this research can be found in (Kazemi et al., 2020). Dynamic graph learning aims to encapsulate the temporal dynamics of graphs and continually refine graph representations, leveraging access to all historical data. In contrast, GIL grapples with the challenge of catastrophic forgetting, where previous task data is either inaccessible or constrained. During evaluations, dynamic graph learning algorithms typically emphasize the latest data, whereas GIL models need to consider historical data as well.

## 6 CONCLUSION

This paper offers a comprehensive theoretical exploration of the learnability of the NGIL problem within the context of evolving graphs. Our formal analysis reveals a crucial insight: NGIL may not be always learnable when structural shifts are uncontrolled. Furthermore, we introduce a novel experience replay framework, SEA-ER, adept at addressing the challenges of catastrophic forgetting and structural shifts inherent in this problem. Empirical results not only validate our theoretical findings but also underscore the effectiveness of our proposed approach.

### 6.1 LIMITATION AND FUTURE WORK

The impossibility result presented in this paper is framed with the existence of a learning algorithm for NGIL. While it can be interpreted as a lower bound on the extent of a structural shift that can induce catastrophic forgetting, having a more quantitative relation on how structural shift lower bound the catastrophic forgetting in NGIL would be a valuable avenue for further research. This would further enhance our understanding of NGIL and facilitate the development of improved methods. However, such quantitative analysis lies beyond the scope of this paper and represents an interesting direction for future work.

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

# A   FURTHER DISCUSSION ON LEARNABILITY AND PROOF OF THE THEOREM 2.3

In this appendix, we provide a proof for Theorem 2.3. The key idea of the proof is to establish a formal connection between the NGIL-2 problem with the domain adaptation problem (henceforth we refer to as the DA problem). We begin by giving a brief introduction to domain adaptation (DA) problem. Without loss of generality, we focus the analysis on binary classification task. Obviously, if binary classification is impossible, then an arbitrary k-way classification is also impossible.

## A.1   DOMAIN ADAPTATION

Let $\mathcal{X}$ and $\mathcal{Y} = \{0, 1\}$ be the input and output space, and $l : \mathcal{X} \mapsto \mathcal{Y}$ be the labelling function. In the DA problem, there are two different distributions over $\mathcal{X}$, the source distribution $\mathcal{S}$ and the target distribution $\mathcal{T}$. The goal of domain adaptation is to learn a classifier $h : \mathcal{X} \mapsto \mathcal{Y}$ that performs well on the target distribution, i.e., $R_{\mathcal{T}}^l(h)$ is small, with the knowledge from the source domain $\mathcal{S}$. Similar to the NGIL-2 problem, the learnability of DA problem can be defined as follow.

**Definition A.1** (DA-learnability). Let $\mathcal{S}$ and $\mathcal{T}$ be distribution over $\mathcal{X}$, $\mathcal{H}$ a hypothesis class, $\epsilon > 0, \delta > 0$. We say that the DA problem is learnable if there exists a learning algorithm that produces a classifier $h$ from $\mathcal{H}$ with probability at least $1 - \delta$, and $h$ incurs an error smaller than $\epsilon$, i.e., $Pr[R_{\mathcal{T}}^l(h) \leq \epsilon] \geq 1 - \delta$, when given access to labelled samples $L$ of size $m$ from $\mathcal{S}$ and unlabelled samples $U$ of size $n$ from $\mathcal{T}$.

## A.2   CONNECTION BETWEEN DA AND NGIL-2

Recall that The NGIL-2 problem is characterized by three distributions: $\mathbb{P}(\mathbf{y}, \mathbf{g}_v | \mathcal{V}_1, \mathcal{G}_{\tau_1})$, $\mathbb{P}(\mathbf{y}, \mathbf{g}_v | \mathcal{V}_1, \mathcal{G}_{\tau_2})$, and $\mathbb{P}(\mathbf{y}, \mathbf{g}_v | \mathcal{V}_2, \mathcal{G}_{\tau_2})$. To simplify the notation, we re-write $\mathbb{P}(\mathbf{y}, \mathbf{g}_v | \mathcal{V}_1, \mathcal{G}_{\tau_1})$, $\mathbb{P}(\mathbf{y}, \mathbf{g}_v | \mathcal{V}_1, \mathcal{G}_{\tau_2}), \mathbb{P}(\mathbf{y}, \mathbf{g}_v | \mathcal{V}_2, \mathcal{G}_{\tau_2})$ as $\mathbb{P}_1^{(1)}, \mathbb{P}_1^{(2)}, \mathbb{P}_2^{(2)}$.

Then, the learnability of NIGL-2 problem can be formally defined as,

**Definition A.2** (NGIL-2-learnability). Given a NGIL-2 with distributions $\mathbb{P}_1^{(1)}, \mathbb{P}_1^{(2)}, \mathbb{P}_2^{(2)}$ and a labelling function $l$, let $\mathcal{H}$ a hypothesis class, $\epsilon > 0, \delta > 0$. We say that the NGIL-2 problem is learnable relative to $\mathcal{H}$, if there exists a learning algorithm that outputs a classifier $h$ from $\mathcal{H}$ with probability of at least $1 - \delta$ and $h$ has error less than $\epsilon$, i.e., $Pr[R_{\mathbb{P}}^l(h) \leq \epsilon] \geq 1 - \delta$, when given access to labeled samples $L$ from $\mathbb{P}$ ($\mathbb{P}$ is the mixed distribution of $\mathbb{P}_1^{(1)}$ and $\mathbb{P}_2^{(2)}$) of size $m + n$ and unlabelled samples $U$ of size $k$ from $\mathbb{P}_1^{(2)}$

To establish a formal connection between NGIL-2 and DA, we consider a relaxed version of the NGIL-2 problem where the learning algorithm allow to access the same amount of data in an arbitrary order from distribution $\mathbb{P}_1^{(1)}$ and $\mathbb{P}_2^{(2)}$ and only needs to performance well on the updated distribution $\mathbb{P}_1^{(2)}$. We refer to this relaxed version of NGIL-2 problem as NGIL-2-weak and its learnability is defined as follows.

**Definition A.3** (NGIL-2-weak-learnability). Given a NGIL-2 with distributions $\mathbb{P}_1^{(1)}, \mathbb{P}_1^{(2)}, \mathbb{P}_2^{(2)}$ and a labelling function $l$, let $\mathcal{H}$ a hypothesis class, $\epsilon > 0, \delta > 0$. We say that the NGIL-2 problem is learnable relative to $\mathcal{H}$, if there exists a learning algorithm that outputs a classifier $h$ from $\mathcal{H}$ with probability of at least $1 - \delta$ and $h$ has error less than $\epsilon$, i.e., $Pr[R_{\mathbb{P}_1^{(2)}}^l(h) \leq \epsilon] \geq 1 - \delta$, when given access to labeled samples $L$ from $P$ ($P$ is the mixed distribution of $\mathbb{P}_1^{(1)}$ and $\mathbb{P}_2^{(2)}$) of size $m + n$ and unlabelled samples $U$ of size $k$ from $\mathbb{P}_1^{(2)}$

It is obvious that the NGIL-2-weak-learnability definition subsume the NGIL-2-learnability. In other words, we can obtain the following corollary immediately from the definition.

**Corollary A.4.** *If the NGIL-2-weak problem is not learnable, then the NGIL-2 problem is not learnable.*

Next, we establish a formal connection between the DA problem and the NGIL-2-weak problem.

**Theorem A.5** (Connection of DA and NGIL-2-weak). *Let $\mathcal{X}$ be a domain and $\mathcal{H}$ the hypothesis class on $\mathcal{X} \times \{0,1\}$. Suppose an instance of the DA problem with distributions $\mathcal{S}$ and $\mathcal{T}$ on $\mathcal{X}$, and there is access to $m'$ labelled samples from $\mathcal{S}$ and $k'$ unlabelled samples from $\mathcal{T}$. Similarly, suppose an instance of an NGIL-2-weak problem with distributions $\mathbb{P}_1^{(1)}, \mathbb{P}_1^{(2)}, \mathbb{P}_1^{(2)}$ over $\mathcal{X}$,and there is access to $m + n$ labelled samples from $\mathbb{P}_1^{(1)}, \mathbb{P}_1^{(2)}$ and $k$ unlabelled samples from $\mathbb{P}_1^{(2)}$. For the same hypothesis sapce $\mathcal{H}$, assuming that $m' = m + n$, $k' = k$, $\lambda_{\mathcal{H}}(\{\mathcal{S}, \mathcal{T}\}) = \lambda_{\mathcal{H}}(\{P, \mathbb{P}_1^{(2)}\})$ and $d_{\mathcal{H} \triangle \mathcal{H}}(\mathcal{S}, \mathcal{T}) = d_{\mathcal{H} \triangle \mathcal{H}}(P, \mathbb{P}_1^{(2)})$, then we have that the DA is learnable if and only if the NGIL-2-weak problem is learnable.*

*Proof.* We first show that under the premise of the theorem if the NGIL-2-weak learnable, then the DA problem is learnable.

Suppose we are given an instance of DA problem of distributions $\mathcal{S}, \mathcal{T}$ and a $\mathcal{H}$. Let $\mathcal{S}$ be a mixed distribution of $\mathcal{S}_1$ and $\mathcal{S}_2$. Then, we can map the DA to a NGIL-2-weak problem as follows.

$$\mathcal{S}_1 = \mathbb{P}_1^{(1)}, \mathcal{S}_2 = \mathbb{P}_1^{(2)}, \mathcal{T} = \mathbb{P}_1^{(2)}$$

Under the mapping above, we have $d_{\mathcal{H} \triangle \mathcal{H}}(\mathcal{S}, \mathcal{T}) = d_{\mathcal{H} \triangle \mathcal{H}}(P, \mathbb{P}_1^{(2)})$ and $\lambda_{\mathcal{H}}(\{\mathcal{S}, \mathcal{T}\}) = \lambda_{\mathcal{H}}(\{P, \mathbb{P}_1^{(2)}\})$, where $P$ is the mixed distribution of distributions $\mathbb{P}_1^{(1)}$ and $\mathbb{P}_1^{(2)}$.

By the assumption, the NGIL-2-weak problem is learnable. By Def. A.3, this means that there exist a learning algorithm that takes in $m + n$ labelled data from $P$ and $k$ unlabelled data from $\mathbb{P}_1^{(2)}$, and with probability at least $1 - \delta$, outputs a classifier $h$ wiht at most $\epsilon$ error with respect to $\mathbb{P}_1^{(2)}$. By the mapping above and the premise of the theorem that $m + n = m'$, $k' = k$, the same learning algorithm would satisfy Def. A.1. This shows that the learnability of NGIL-2-weak implies the learnability of DA problem.

Now we show the opposite that under the premise of the theorem if the DA problem is learnable, then the NGIL-2-weak learnable.

Let's reverse the construction direction. Suppose we are given an instance of NGIL-2-weak problem of distributions $\mathbb{P}_1^{(1)}, \mathbb{P}_1^{(2)}, \mathbb{P}_1^{(2)}$ and a $\mathcal{H}$. Let $\mathcal{P}$ be a mixed distribution of $\mathbb{P}_1^{(1)}$ and $\mathbb{P}_1^{(2)}$. Then, we can map the NGIL-2-weak problem to DA problem as follows.

$$P = \mathcal{S}, \mathbb{P}_1^{(2)} = \mathcal{T}$$

Similarly, under the mapping above, we have $d_{\mathcal{H} \triangle \mathcal{H}}(\mathcal{S}, \mathcal{T}) = d_{\mathcal{H} \triangle \mathcal{H}}(P, \mathbb{P}_1^{(2)})$ and $\lambda_{\mathcal{H}}(\{\mathcal{S}, \mathcal{T}\}) = \lambda_{\mathcal{H}}(\{P, \mathbb{P}_1^{(2)}\})$.

By the assumption, the DA problem is learnable. By Def. A.1, this means that there exist a learning algorithm that takes in $m'$ labelled data from $\mathcal{S}$ and $k'$ unlabelled data from $\mathcal{T}$, and with probability at least $1 - \delta$, outputs a classifier $h$ wiht at most $\epsilon$ error with respect to $\mathcal{T}$. By the mapping above and the premise of the theorem that $m + n = m'$, $k' = k$, the same learning algorithm would satisfy Def. A.3. This shows that the learnability of DA problem implies the learnability of the NGIL-2-weak problem.

This completes the proof of Theorem A.5. $\qquad\square$

Now that we have shown that the learnability of DA problem is equivalent to the learnability of the NGIL-2-weak problem. Combining Theorem A.5 and Corollary A.4, we immediately obtain the following corollary on the connection between the NGIL-2 problem with Def. A.2 and the domain adaptation problem.

**Corollary A.6** (Connection of DA and NGIL-2). *Let $\mathcal{X}$ be a domain and $\mathcal{H}$ the hypothesis class on $\mathcal{X} \times \{0,1\}$. Suppose an instance of the DA problem with distributions $\mathcal{S}$ and $\mathcal{T}$ on $\mathcal{X}$, and there is access to $m'$ labelled samples from $\mathcal{S}$ and $k'$ unlabelled samples from $\mathcal{T}$. Similarly, suppose an instance of a NGIL-2 problem with distributions $\mathbb{P}_1^{(1)}, \mathbb{P}_1^{(2)}, \mathbb{P}_1^{(2)}$ over $\mathcal{X}$,and there is sequential access to $m$ labelled samples from $\mathbb{P}_1^{(1)}$, $n$ labelled samples from $\mathbb{P}_1^{(2)}$ and $k$ unlabelled samples from $\mathbb{P}_1^{(2)}$. For the same hypothesis sapce $\mathcal{H}$, assuming that $m' = m + n$, $k' = k$, $\lambda_{\mathcal{H}}(\{\mathcal{S}, \mathcal{T}\}) =$*

$\lambda_{\mathcal{H}}(\{P, \mathbb{P}_1^{(2)}\})$ and $d_{\mathcal{H}\Delta\mathcal{H}}(\mathcal{S}, \mathcal{T}) = d_{\mathcal{H}\Delta\mathcal{H}}(P, \mathbb{P}_1^{(2)})$, then we have that the if the DA is not learnable, then the NGIL-2 problem is not learnable.

### A.3 PROOF FOR THEOREM 2.3

Corollary A.6 above establishes a formal connection between the DA problem and the NGIL-2 problem. Under the same proposed mapping, the infeasibility of DA implies the infeasibility of NGIL-2. DA is a well-studied problem with mature understanding on its learnability.

David et al. (2010) has shown that small divergent distance between source domain and target domain is a "necessary" condition for the success of domain adaptation. We restate the theorem below.

**Theorem A.7** ( (David et al., 2010)). *Let $\mathcal{X}$ be some domain set, and $\mathcal{H}$ a class of functions over $\mathcal{X} \times \{0,1\}$. For every $c > 0$ there exists probability distributions $\mathcal{S}, \mathcal{T}$ over $\mathcal{X}$ such that for every domain adaptation learner, every integers $m, n > 0$, there exists a labeling function $l : \mathcal{X} \mapsto 0, 1$ such that $\lambda_{\mathcal{H}}(P, Q) \leq c$ and the DA problem is not learnable.*

The proof of Theorem A.7 can be found in (David et al., 2010). Then the proof of Theorem 2.3 follows immediately from Theorem A.7 and Corollary A.6.

## B EXPRESSIVENESS AND SAMPLE COMPLEXITY

In this appendix, building upon the connection established in the previous appendix, we provide a further discussion of the sample complexity and expressiveness.

**Theorem B.1** (Necessity of the Expressiveness of Hypothesis Space). *Let $\mathcal{X}$ be some domain set, and $\mathcal{H}$ be a class of functions over $\mathcal{X} \times \{0,1\}$ whose VC dimension is much smaller than $|\mathcal{X}|$. Then for every $c > 0$, there exists a NGIL-2 problem such that $d_{\mathcal{H}\Delta\mathcal{H}}(\mathbb{P}, \mathbb{P}_1^{(2)}) \leq c$, where $\mathbb{P}$ is the mixed distribution of $\mathbb{P}_1^{(1)}$ and $\mathbb{P}_2^{(2)}$, and the NGIL-2 problem is not learnable with any amount of labelled data from $\mathbb{P}_1^{(1)}, \mathbb{P}_2^{(2)}$ and unlablled data from $\mathbb{P}_1^{(2)}$.*

Theorem B.1 formalizes the intuition that the hypothesis class (GNN model) needs to be expressive enough to capture different distributions in NGIL, for it to be learnable.

**Theorem B.2** (Sample Complexity). *Let $\mathcal{X}$ be some compact domain set in $\mathbf{R}^d$, and $\mathcal{H}$ be a class of functions over $\mathcal{X} \times \{0,1\}$. Suppose the labeling function $l : \mathcal{X} \mapsto \{0,1\}$ is $\alpha$-Lipschitz continuous. Then, if $m + n + k < \sqrt{(1 - 2(\epsilon + \delta))(\alpha + 1)^d}$, for every $c > 0$, there exists a NGIL-2 problem such that $\lambda_{\mathcal{H}}(\{\mathbb{P}_1^{(1)}, \mathbb{P}_2^{(2)}, \mathbb{P}_1^{(2)}\}) \leq c$, $d_{\mathcal{H}\Delta\mathcal{H}}(P, \mathbb{P}_1^{(2)}) \leq c$, where $\mathbb{P}$ is the mixed distribution of $\mathbb{P}_1^{(1)}$ and $\mathbb{P}_2^{(2)}$, and the NGIL-2 problem is not learnable.*

Theorem B.2 suggests that even if the two "necessary" conditions in Theorem 2.3 and Theorem B.1 are satisfied, there is still a sampling requirement depending on the dimension of the input domain and the smoothness of the labelling function. This shows the importance of having access to previous data for NGIL.

### B.1 PROOF FOR THEOREM B.1 AND THEOREM B.2

The idea of the proof is similar to the one presented earlier. (David et al., 2010) has shown that expressive power of the hypothesis space is a "necessary" condition for the success of domain adaptation. We restate the theorem below.

**Theorem B.3.** *Let $\mathcal{X}$ be some domain set, and $\mathcal{H}$ a class of functions over $\mathcal{X} \times \{0,1\}$ whose VC dimension is much smaller than $|\mathcal{X}|$. Then for every $c > 0$ there exists probability distributions $Q, P$ over $\mathcal{X}$ such that for every domain adaptation learner, every integers $m, n > 0$, there exists a labeling function $l : \mathcal{X} \mapsto 0, 1$ such that $d_{\mathcal{H}\Delta\mathcal{H}}(P, Q) \leq c$ and the DA problem is not learnable.*

The proof of Theorem B.3 can be found in (David et al., 2010). Then the proof of Theorem B.1 follows immediately from Theorem B.3 and Corollary A.6.

(Ben-David & Urner, 2012) has shown that even if DA problem has small divergent distance and the hypothesis space is expressive enough, there is still a sampling requirement for the DA problem to be learnable. We restate the theorem below.

**Theorem B.4.** *Let $\mathcal{X} \subset \mathbf{R}^d$ and $\mathcal{H}$ a class of functions over $\mathcal{X} \times \{0,1\}$. Suppose the labelling function $l$ is $\alpha$-Lipschitz continuous. For every $c > 0$ there exist $\mathcal{S}, \mathcal{T}$ over $\mathcal{X}$ suc htat $\lambda_{\mathcal{H}} \leq c$ and $d_{\mathcal{H}\Delta\mathcal{H}}(\mathcal{S}, \mathcal{T}) \leq c$ and the DA problem is not learnable if $m + n \geq \sqrt{(\alpha+1)^d(1 - 2(\epsilon + \delta))}$, where $m, n, \epsilon, \delta$ are given in Def. A.1.*

The proof of Theorem B.4 can be found in (Ben-David & Urner, 2012). Then the proof of Theorem B.2 follows immediately from Theorem B.4 and Corollary A.6.

*Remark* B.5. In addition to the learnability studies, there exist a large body of studies investigating different techniques for domain adaptation. The formal connection A.6 allow to rigorously connection the NGIL-2 problem and DA problem and to carefully transfer or explore the understanding and techniques from DA to NGIL-2.

## C PROOF OF PROPOSITION 3.1

In this appendix, we provide proof for Proposition 3.1. We begin with restating the definition of distortion.

**Definition C.1** (distortion rate). Given two metric spaces $(\mathcal{Q}, d)$ and $(\mathcal{Q}', d')$ and a mapping $\gamma : \mathcal{Q} \mapsto \mathcal{Q}'$, $\gamma$ is said to have a distortion $\alpha \geq 1$, if there exists a constant $r > 0$ such that $\forall u, v \in \mathcal{E}$, $rd(u,v) \leq d'(\gamma(u), \gamma(v)) \leq \alpha rd(u,v)$.

*Proof.* Suppose $P_i \subset \mathcal{V}_i$ is the experience replay set selected by solving Eq. 3. Let $s$ be the structure of interest with distance measure $d_s$. Let $g$ be a given GNN model with distortion $\alpha$ and scaling factor $r$ and its prediction function $g$.

Let $\sigma : \mathcal{V}_i \mapsto P_i$ denote a mapping that map a vertex $v$ from $\mathcal{V}_i$ to the closest vertex in $P_i$, i.e.,

$$\sigma(v) = \arg\min_{u \in P_i} d_s(u, v). \tag{7}$$

Let's consider the loss of vertex $v$, $\mathcal{L}(g \circ g(v), y_v)$. By assumption, $\mathcal{L}$ is smooth and let's denote $B_{\mathcal{L}}^{up1}$ to be the upper bound for the first derivative with respect to $g \circ g(v)$ and $B_{\mathcal{L}}^{up2}$ to be the upper bound for the first derivative with respect to $y_v$. Then, consider the Taylor expansion of $\mathcal{L}(g \circ g(v), y_v)$ with respect to $\sigma(v)$, which is given as follows

$$\begin{aligned} \mathcal{L}(g \circ g(v), y_v) \leq \mathcal{L}(g \circ g(\sigma(v)), y_{\sigma(v)}) + \\ B_{\mathcal{L}}^{up1} \|g \circ g(v) - g \circ g(\sigma(v))\| \\ + B_{\mathcal{L}}^{up2} \|y_v - y_{\sigma(v)}\| \end{aligned} \tag{8}$$

Next, let's examine the inequality above term by term and start with $\|g \circ g(v) - g \circ g(\sigma(v))\|$. Let's denote $h_v$ and $h_{\sigma(v)}$ the embedding for vertex $v$ and $\sigma(v)$. Then, we have that,

$$\|g \circ g(v) - g \circ g(\sigma(v))\| = \|g(h_v) - g(h_{\sigma(v)})\| \tag{9}$$

By definition of distortion as given in Def. C.1, we have that,

$$\|h_v - h_{\sigma(v)}\| \leq r\alpha d_s(v, \sigma(v)) \tag{10}$$

By assumption, the prediction function $g$ is smooth. Let $B_g$ denote the upper bound of first derivative of the prediction function $g$. Then, we have

$$\|g(h_v) - g(h_{\sigma(v)})\| \leq B_g^{up} \|h_v - h_{\sigma(v)}\| \tag{11}$$

Substitute all these back, we have

$$\|g \circ g(v) - g \circ g(\sigma(v))\| \leq B_g^{up} r\alpha d_s(v, \sigma(v)) \tag{12}$$

Next, let's consider $\|y_v - y_{\sigma(v)}\|$. Similarly, by the data smoothness assumption and distortion, we have that

$$\|y_v - y_{\sigma(v)}\| \leq B_l \|h_v - h_{\sigma(v)}\| \leq B_l^{up} r\alpha d_s(v, \sigma(v)) \tag{13}$$

Substitute these back to the inequality we start with, we have that

$$
\begin{aligned}
\mathcal{L}(g \circ g(v), y_v) \leq\ & \mathcal{L}(g \circ g(\sigma(v)), y_{\sigma(v)}) + \\
& B_{\mathcal{L}}^{up1} B_g^{up} r\alpha d_s(v, \sigma(v)) + B_{\mathcal{L}}^{up2} B_l^{up} r\alpha d_s(v, \sigma(v)) \\
\leq\ & \mathcal{L}(g \circ g(\sigma(v)), y_{\sigma(v)}) + \\
& (B_l^{up} B_{\mathcal{L}}^{up1} + B_g^{up} B_{\mathcal{L}}^{up2}) r\alpha \max_{v' \in \mathcal{V}_i} d_s(v', \sigma(v')) \\
=\ & \mathcal{L}(g \circ g(\sigma(v)), y_{\sigma(v)}) + \\
& (B_l^{up} B_{\mathcal{L}}^{up1} + B_g^l B_{\mathcal{L}}^{up2}) r\alpha D_s(\mathcal{V}_i, P_i).
\end{aligned}
\tag{14}
$$

Since the inequality above holds for every vertex $v$, then we have

$$
\begin{aligned}
R^{\mathcal{L}}(g \circ g, \mathcal{V}_i) \leq\ & R^{\mathcal{L}}(g \circ g, P_i) + \\
& (B_l^{up} B_{\mathcal{L}}^{up1} + B_g^{up} B_{\mathcal{L}}^{up2}) r\alpha D_s(\mathcal{V}_i, P_i) \\
=\ & R^{\mathcal{L}}(g \circ g, P_i) + \mathcal{O}(\alpha D_s(\mathcal{V}_i, P_i))
\end{aligned}
\tag{15}
$$

$\square$

# D    ADDITIONAL EXPERIMENTAL DETAILS

In this appendix, we provide additional experimental results and include a detailed set-up of the experiments for reproducibility.

## D.1    HARDWARE AND SYSTEM

All the experiments of this paper are conducted on the following machine

CPU: two Intel Xeon Gold 6230 2.1G, 20C/40T, 10.4GT/s, 27.5M Cache, Turbo, HT (125W) DDR4-2933

GPU: four NVIDIA Tesla V100 SXM2 32G GPU Accelerator for NV Link

Memory: 256GB (8 x 32GB) RDIMM, 3200MT/s, Dual Rank

OS: Ubuntu 18.04LTS

## D.2    DATASET AND PROCESSING

### D.2.1    DATASET DESCRIPTION

**OGB-Arxiv.** The OGB-Arxiv dataset Hu et al. (2020) is a benchmark dataset for node classification. It is constructed from the arXiv e-print repository, a popular platform for researchers to share their preprints. The graph structure is constructed by connecting papers that cite each other. The node features include the text of the paper's abstract, title, and its authors' names. Each node is assigned one of 40 classes, which corresponds to the paper's main subject area.

**Cora-Full.** The Cora-Full Bojchevski & Günnemann (2017) is a benchmark dataset for node classification. Similarly to OGB-Arxiv, it is a citation network consisting of 70 classes.

**Reddit.** The Reddit dataset Hamilton et al. (2017) is a benchmark dataset for node classification that consists of posts and comments from the website Reddit.com. Each node represents a post or comment and each edge represents a reply relationship between posts or comments.

### D.2.2    LICENSE

The datasets used in this paper are curated from existing public data sources and follow their licenses. OGB-Arxiv is licensed under Open Data Commons Attribution License (ODC-BY). Cora-Full dataset and the Reddit dataset are two datasets built from publicly available sources (public papers and Reddit posts) without a license attached by the authors.

Table 3: Incremental learning settings for each dataset.

| Datasets | OGB-Arxiv | Reddit | CoraFull |
|---|---|---|---|
| # vertices | 169,343 | 227,853 | 19,793 |
| # edges | 1,166,243 | 114,615,892 | 130,622 |
| # class | 40 | 40 | 70 |
| # tasks | 20 | 20 | 35 |
| # vertices / # task | 8,467 | 11,393 | 660 |
| # edges / # task | 58,312 | 5,730,794 | 4,354 |

### D.2.3 DATA PROCESSING

For the datasets, we remove the 41-th class of Reddit-CL, following closely in Zhang et al. (2022). This aims to ensure an even number of classes for each dataset to be divided into a sequence of 2-class tasks. For all the datasets, the train-validation-test splitting ratios are 60%, 20%, and 20%. The train-validation-test splitting is obtained by random sampling, therefore the performance may be slightly different with splittings from different rounds of random sampling.

### D.3 INCREMENTAL TRAINING PROCEDURE

Suppose $\tau_1, \tau_2, ...\tau_m$ is the set of classification tasks created from the dataset. Then, the standard incremental training procedure is given in Alg. 1 and a graphical illustration is given in Fig. 7.

### D.4 CONTEXTUAL STOCHASTIC BLOCK MODEL

We use the contextual stochastic block model (Deshpande et al., 2018) to create a graph for each learning stage. For each graph, we create 300 vertices and assign them a community label $\{0, 1\}$. We refer to a specific ratio of vertices from different communities as a configuration and the formula we use to vary the community ratio is as follows, configuration $i = (150 + (i - 1) * 25, 150 - (i - 1) * 25, 150 - (i - 1) * 25, 150 + (i - 1) * 25)$. As $i$ increases, the difference in the ratio of the two learning stages increases. Intuitively, this should allow us to control the distribution difference between different learning stages.

Then we follow the standard stochastic block model. We use $p_{intra} = 0.15$ to randomly create edges among vertices of the same community and $p_{inter} = 0.1$ to randomly create edges among vertices of different communities. Then, we use $p_{stage} = [0.025, 0.05, 0.08, 0.1, 0.15]$ to create edges among vertices of different learning stages.

Then, the node features $x_v$ of each vertex is created based on the following formula,

$$x_v = \sqrt{\frac{\mu}{n}} y_v u + \frac{Z_v}{p},$$

where $y_v \in \{0, 1\}$ is the community label of vertex $v$, $u, Z_v \sim \mathcal{N}(0, I)$ are random Gaussian of dimension $p$, $n$ is the number of vertices in the graph and $\mu$ is a hyper-parameter of the model to control how separable of the features of the vertices from different communities. For our experiments, we use $\mu = 5, n = 300, p = 500$ for the graph in each stage.

### D.5 GRAPH NEURAL NETWORK DETAILS

For our experiments, we use two popular GNN models, GCN (Kipf & Welling, 2017) and Graph-Sage (Hamilton et al., 2018). For implementation, we use the default implementation provided by DGL for realizing these models. For all our experiments, we use two GNN layers followed by an additional readout function of an MLP layer. We use Sigmoid as the activation function for the last hidden layer and Relu for the rest of the layers. We only vary the hidden dimension of the GNN layer for each experiment.

## D.6  Difference between Inductive and Transductive

In this subsection, we provide the additional results on the remaining dataset for the setting with evolving graphs and constant graphs. The results are reported in Fig. 5 and . **??**.

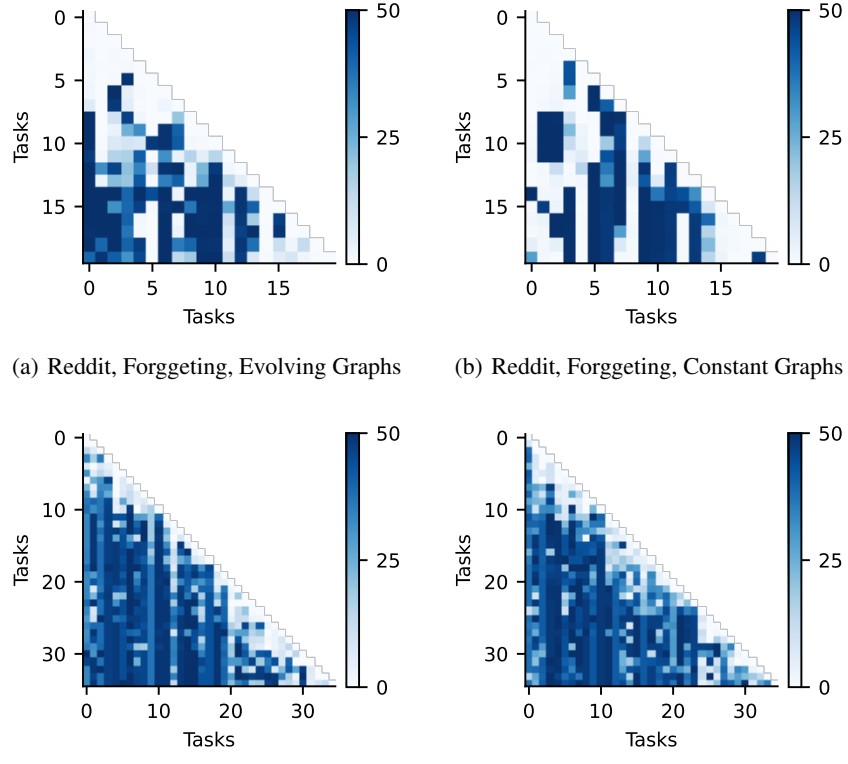

(a) Reddit, Forggeting, Evolving Graphs  (b) Reddit, Forggeting, Constant Graphs

(c) CoraFull, Forggeting, Evolving Graphs  (d) CoraFull, Forggeting, Constant Graphs

Figure 4: Performance Matrix of Bare Model on Transductive and Inductive Setting on CoraFull and Reddit Datasets.

---

**Algorithm 1** Incremental Training

---

**Input:** $\tau_1, \tau_2, ...\tau_m$ //classification task created from the dataset.
For $\tau$ in $\tau_1, \tau_2, ...\tau_m$:

1. update the graph structure with data from $\tau_i$
2. get $V_i^{\text{train}}$ for $\tau_i$
3. update the existing GNN model and train a new task prediction head on $V_i^{\text{train}}$
4. if evaluation, evaluate the model on the current and previous task

---

# E  Related Works

In this appendix, we present a more detailed discussion of the related works as complementary to the related work section in the main papers.

## E.1  Incremental Learning

Incremental learning, also known as continual or lifelong learning, has garnered increased attention in recent years and has been extensively investigated for Euclidean data. For a more comprehensive review of these works, we direct readers to the surveys (Delange et al., 2021; Parisi et al., 2019; Biesialska et al., 2020). The primary challenge in incremental learning lies in addressing the

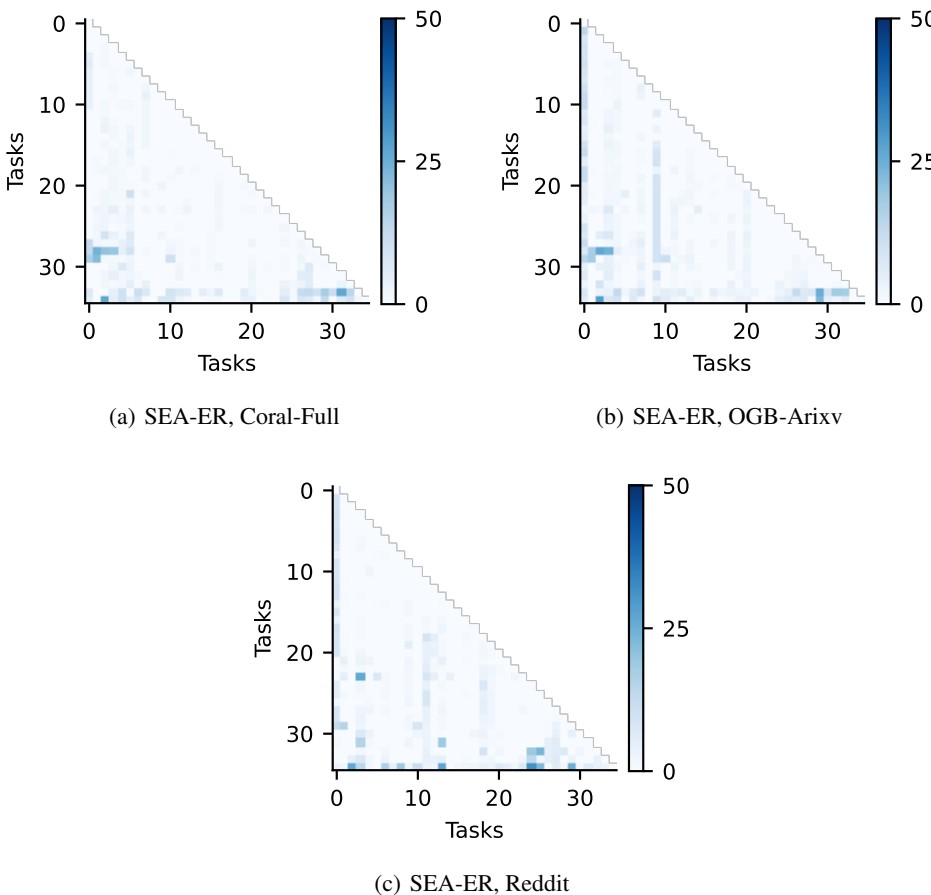

(a) SEA-ER, Coral-Full          (b) SEA-ER, OGB-Arixv

(c) SEA-ER, Reddit

Figure 5: Forgetting Performance Matrix of SEA-ER on CoraFull and Reddit Datasets.

catastrophic forgetting problem, which is the significant decline in the performance of a model on previous tasks after training on new tasks.

Current approaches to mitigate this problem can be broadly classified into three categories: regularization-based methods, experience-replay-based methods, and parameter-isolation-based methods. Regularization-based methods aim to preserve the performance of models on previous tasks by penalizing substantial changes in the model parameters (Jung et al., 2016; Li & Hoiem, 2017; Kirkpatrick et al., 2017; Farajtabar et al., 2020; Saha et al., 2021). Parameter-isolation-based methods avert drastic alterations to the parameters crucial for previous tasks by consistently introducing new parameters for new tasks (Rusu et al., 2016; Yoon et al., 2017; 2019; Wortsman et al., 2020; Wu et al., 2019). Experience-replay-based methods select a set of representative data from previous tasks, which are then used to retrain the model alongside new task data, preventing forgetting (Lopez-Paz & Ranzato, 2017; Shin et al., 2017; Aljundi et al., 2019; Caccia et al., 2020; Chrysakis & Moens, 2020; Knoblauch et al., 2020). Given the distinctive challenges of NGIL, particularly structural shifts, these established techniques may not be directly applicable. Hence, in this study, we introduce a novel experience replay method specifically crafted for NGIL.

## E.2 INCREMENTAL LEARNING WITH GRAPH NEURAL NETWORKS

Recent strides in GNNs have stimulated increased exploration in GIL owing to its pragmatic relevance (Wang et al., 2022; Xu et al., 2020; Daruna et al., 2021; Kou et al., 2020; Ahrabian et al., 2021; Cai et al., 2022; Wang et al., 2020; Liu et al., 2021; Zhang et al., 2021; Zhou & Cao, 2021b; Carta et al., 2021; Kim et al., 2022; Su et al., 2023). For a detailed examination of GIL methodologies and

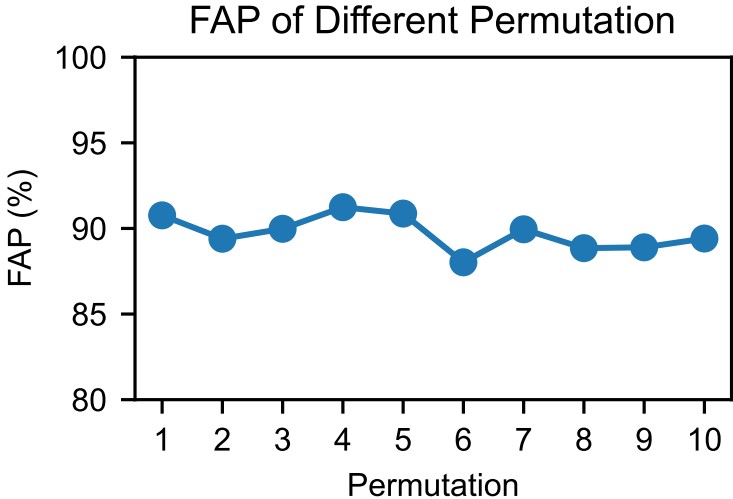

Figure 6: FAP performance of our method with respect to different permutations. X-axis indicate different permutation 1-10. The figure shows that different permutations have a small effect ( 1%) on the performance.

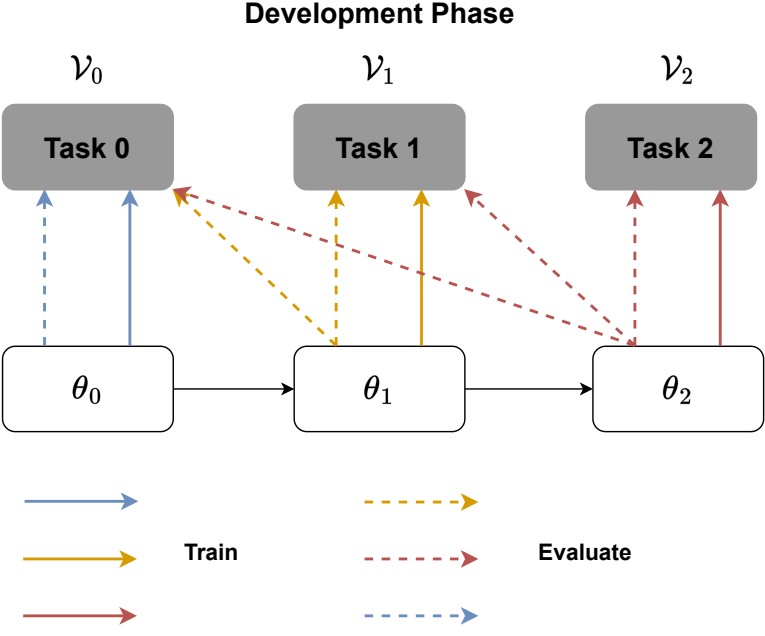

Figure 7: Incremental Training in Development Phase. The figure above illustrates what task data the model use for training and evaluation.

their efficacy, we direct readers to recent reviews and benchmarks (Zhang et al., 2022; Yuan et al., 2023; Febrinanto et al., 2023a).

Studies that are closely related to this work include (Ahrabian et al., 2021; Zhou & Cao, 2021b; Kim et al., 2022; Su et al., 2023). (Zhou & Cao, 2021b) has shown the feasibility and effectiveness of the experience replay framework in addressing the catastrophic forgetting problem in NGIL. However, they focus on a transductive setting where the complete graph structure is available before training and their experience replay method has neglected the structural shift problem. (Kim et al., 2022)

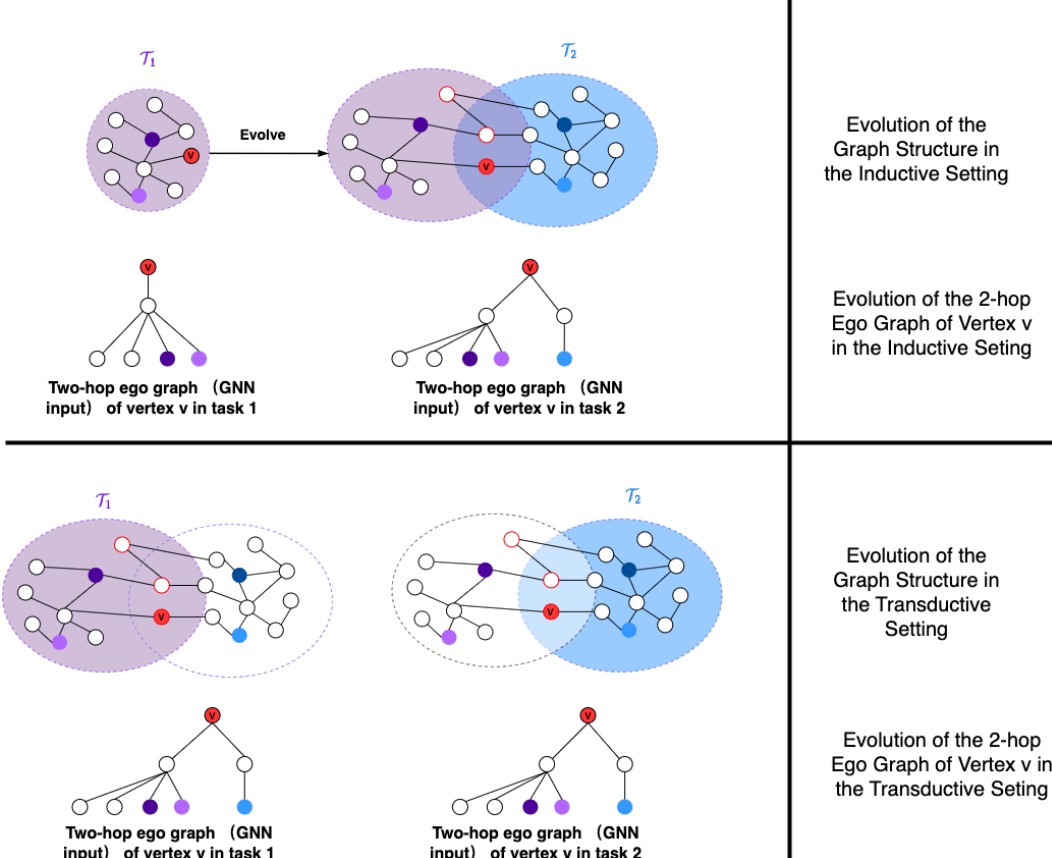

Figure 8: Difference between Inductive (evolving graphs) GIL and Transductive (constant graphs) GIL. As illustrated in the top half of the figure, in the evolving graph setting, the appearing tasks expand/change the existing graph structure, resulting in changes in the neighborhood (input to the GNN) of some previous vertices. As discussed in the main body of the paper, this structural changes/dependency would cause distribution shift in the previously learnt information of the model. On the other hand, in the constant graph setting, because the complete graph structure is available from te very first beginning. The neighborhoods of all the vertices are static/stay unchanged throughout different learning stages. Therefore, structural dependency is an unique challenge in the incremental learning in evolving graph.

further investigates the changing receptive field induced by evolving graph structures using a full-graph formulation (which does not scale well as graphs grow) and proposes an experience replay strategy based on the influence score of node representation. Meanwhile, (Ahrabian et al., 2021) takes into account the topological awareness of GNNs and proposes an experience buffer selection based on the node degree.

There are three key distinctions between our work and these three studies. First, as discussed in (Zhang et al., 2022), both (Zhou & Cao, 2021b) and (Ahrabian et al., 2021) are restricted to a transductive setting, where the underlying graph structure is assumed to be given or independent throughout the learning process. This assumption neglects the structural shift issue of graph structure when transitioning to a new task, which requires specific treatment. As shown in (Su et al., 2023), structural shift induced by changing structure can seriously affect the performance of the model. Second, our proposed experience buffer selection relies on the topological awareness of GNNs and offers a theoretical guarantee that ensures consistent long-term performance. Lastly, instead of simply blending the experience buffer with the training set (Ahrabian et al., 2021; Zhou & Cao, 2021b; Kim et al., 2022), we propose a novel replay method with importance reweighting for addressing the structural shift problem.

Therefore, the inductive NGIL, wherein the graph evolves with the introduction of new tasks, remains relatively uncharted and demands a robust theoretical framework. Su et al. (2023) provides the first formal study into the structural shift issue of inductive NGIL. Nevertheless, their research focuses on establishing a relation between structural shift and catastrophic forgetting in NGIL. Our works can be seen as an important extension and endeavours to further fill the void of theoretical studies of NGIL by presenting the first impossibility result on the learnability of the NGIL problem, highlighting the pivotal role of the structural shift.

Lastly, it is important to mention another research stream called dynamic/temporal graph learning, focusing on GNNs adept at capturing changing graph structures. A comprehensive review of this can be found in (Kazemi et al., 2020). Dynamic graph learning aims to encapsulate the graph's temporal dynamics and persistently refine graph representations, with the entirety of past data at its disposal. Conversely, GIL grapples with the catastrophic forgetting dilemma, and previous task data is either inaccessible or restricted. In addition, during evaluations, a dynamic graph learning algorithm focuses on the latest data, whereas GIL models account for historical data as well.

## F COMPLETE FRAMEWORK AND PESUDO-CODE

### F.1 OVERALL FRAMEWORK

The incremental learning procedure of our framework is summarized in Alg. 2 and Fig. 9 provide a graphical illustration on the training of the framework on a two-stage incremental learning. Selected based on the structural bias of GNNs, the proposed structure-aware replay, as well as the domain distance regularization, can mitigate both structural dependency and sequential bias in NCTIL-EG.

---

**Algorithm 2** SEA-ER-GNN

   **Input:** $\tau_i$ //new node classification task
**Require:** $\mathcal{P}_{i-1} = \{P_1, P_2, ..., P_{i-1}\}$ //replay buffer
**Require:** $g$ //current GNN model
**Require:** $\mathcal{G}_{\tau_{i-1}}$ //graph induced by vertices of $\tau_{i-1}$
**Require:** $\beta_l, \beta_u$
   Create the training set for this learning stage

$$\hat{\mathcal{V}}_i = \mathcal{V}^{\text{train}} \cup \mathcal{P}_{i-1};$$

   Compute $\beta$ with Eq. 6
   Run the training procedure (forward computation and backward propagation) with loss function

$$\mathcal{L} = \frac{1}{|\mathcal{V}_i^{\text{train}}|} \sum_{v \in \mathcal{V}_i^{\text{train}}} \mathcal{L}(v) + \sum_{P_j \in \mathcal{P}_i} \frac{1}{|P_j|} \sum_{v \in P_j} \beta_v \mathcal{L}(v),$$

   Update the replay buffer with replay set $P_i$ from $\tau_i$

$$\mathcal{P}_i = \mathcal{P}_{i-1} \cup P_i;$$

---

### F.2 SELECTING REPLAY SET

The replay set selection strategy is given in Eq. equation 3. It is obvious that when $D_i = V_i$, the problem can be reduced to the k-center problem (Hochbaum, 1996) which is to select a set of k vertices so that the distance from the rest of the vertices to these selected k vertices is minimized. While the k-center problem is NP-hard, there exist efficient greedy approximation algorithms, e.g., by selecting a vertex that is furthest away from the established set (Hochbaum, 1996). However, such a heuristic might run into the problem of selecting vertices that are at the end of the long path and barely connect to other vertices. To mitigate this, we propose to weight the distance with the degree of the vertices. In addition, like other vertex sampling algorithms, we modify the above selection process by assigning a probability to each vertex based on the greedy criteria and its degree to give the algorithm some room for exploration.

Figure 9: Two-stage SEA-ER-GNN Training

---

**Algorithm 3** Replay Set

---

**Input:** $D_i \subset V_i$ //labelled set from task i ($\tau_i$)
**Require:** $\mathcal{G}_{i-1} = \mathcal{G}[\mathcal{V}_{i-1}]$ //existing graph
Sample/select the replay set $P_i$ with the following procedures:

1. initialize each vertice $v \in D_i$ with $p_v$ with its degree and sample an initial vertex $u$ based on a probability proportion to the degree (e.g., normalized each the degree of each vertex with the sum of total degree)
2. assignment $u$ to the replay set $P_i = \{u\}$
3. compute the shortest-path distance between each vertex $v \in D_i \setminus P_i$ to $P_i$ with the formula $d(v, P_i) = \min\{d(v, u)|u \in P_i\}$
4. compute the weight for each vertex $v \in D_i \setminus P_i$ based on product of the distance to $P_i$ and its degree, i.e., $p_v = \text{degree}(v) * d(v, P_i)$
5. sample $v$ from $D_i$ based on probability proportion to $p_v$, e.g., normalized it with the sum from the remaining vertices of $D_i$
6. repeat step 2-5 until $|P_i| = k$

---

### F.3 COMPLEXITY ANALYSIS

The time complexity of our algorithm is readily apparent from its procedural steps, and it exhibits a computational complexity of O(N), where N denotes the number of vertices in the graph. Notably, our approach shares a common characteristic with baseline methods, where the problem formulation might inherently be NP-hard to solve. However, both our method and the baselines leverage efficient

heuristics to obtain practical solutions. As a result, the time complexity of our proposed methods and the baseline approaches remains similar, with all falling within the order of O(N). This underscores the computational efficiency and feasibility of our methodology in addressing the challenges posed by the problem at hand.

