# OpenReview forum: "ON LEARNABILITY AND EXPERIENCE REPLAY METHODS FOR GRAPH INCREMENTAL LEARNING ON EVOLVING GRAPHS"
_ICLR.cc/2024/Conference — Submitted to ICLR 2024_

### Official Review · Reviewer_vspK · 2023-10-28

**Soundness:** 2 fair
**Presentation:** 2 fair
**Contribution:** 2 fair
**Rating:** 5
**Confidence:** 3

**Summary:**

The authors study the problem of graph incremental learning, where (a batch of) nodes arrive at each time step. We hope to update our model efficiently in this setting as in the standard incremental or online learning setting. The authors claim that this problem is not ``learnable`` when the structural shift is not controlled. They propose a replay-based method to mitigate the effect of structural shift.

**Strengths:**

-	The studied problem is important.
-	The extensive experiments show that the proposed method outperforms prior works.

**Weaknesses:**

-	The clarity, especially for the theoretical results, can be improved.
-	The Theorem 2.3 seems problematic. See the counter-example below.
-	It is not clear why the proposed method can mitigate the issue of ``structural shift``.

I really like to convey ideas and it is great to see the authors attempt to provide some analysis to understand the problem of Node-wise Graph Incremental Learning (NGIL). However, I find it hard to understand the impossibility result of NGIL and why the proposed method can solve the issue. For instance, what do the authors mean by a **good** classifier $f$ in Theorem 2.3? What is exactly the setting of available training data information we can use at each time step considered in Theorem 2.3? Clearly, it has to rule out the case of retraining from scratch (denoted as Joint Training by the authors) otherwise Theorem 2.3 makes no sense. Unfortunately, I cannot see where and how Theorem 2.3 rules out this method. Also, the authors assume that we can sample the $k$-hop ego-subgraph for all nodes in $\mathcal{V}_i$ at each time step $i$. Notably, the statement of Theorem 2.3 is independent of the choice of $k$. As a result, if we set $k$ sufficiently large, then getting $g_v$ is equivalent to getting the entire graph. Thus, retraining from scratch is included in this scenario. I feel there must be some other assumption in order to make Theorem 2.3 reasonable, and I hope the authors can state them clearly.

On the other hand, even if the conclusion of Theorem 2.3 is correct the main issue is the structural shift being uncontrolled. The authors claim that their method can mitigate this issue, which sounds weird to me. Note that the structural shift is defined by how a new node (or batch of nodes) is added to the current graph, which changes the graph topology. This is definitely not controllable in practice, as this process depends on the nature of the data but not the algorithm design. I am confused as to why the authors can claim that their method mitigates this issue.

Finally, I wonder how the problem of NGIL is related to graph unlearning [1,2]. If we do not care about the privacy issue in the graph unlearning, essentially NGIL is the reverse direction of graph unlearning. Then for a simple problem and model, the technique in [1,2] seem also provably applicable. I wonder if Theorem 2.3 contradicts the finding of the machine unlearning literature.

## References
[1] Efficient Model Updates for Approximate Unlearning of Graph-Structured Data, Chien et al. ICLR 2023.

[2] Efficiently Forgetting What You Have Learned in Graph Representation Learning via Projection, Cong et al. AISTATS 2023.

**Questions:**

Please check my comments in the weaknesses section. In summary, my questions are:

1.	What does ``good $f$`` mean in Theorem 2.3?

2.	Why Theorem 2.3 is correct? Isn’t retraining from scratch a counter-example?

3.	Since Theorem 2.3 does not depend on the number of hop $k$. If we choose $k$ sufficiently large (i.e., larger than the diameter of the graph), then essentially getting $g_v$ means we get the entire graph, and thus retraining from scratch must be included. Do the authors miss some assumptions for Theorem 2.3 to hold?

4.	Why the proposed method can mitigate the structural shift? Isn’t this dependent on the data nature that we have no way to control?

5. Does Theorem 2.3 contradict with graph unlearning literature [1,2]?

---

> ### Author Response · Authors · 2023-11-16
>
> Thank you for the feedback and comments. Please allow us to address and clarify the concerns as follows:
>
> **Q1. What does `good $f$` mean in Theorem 2.3?**
>
> - In Theorem 2.3, the symbol $\mathcal{F}$, as defined in Section 2.1, represents the hypothesis space of the model, encompassing all possible parameters from the neural network. This is a standard notion employed in statistical learning theory.
>
> **Q2 & Q3. Retraining from scratch as a counter example and potential missing assumption in Theorem 2.3**
>
> Retraining from scratch is NOT a counterexample to Theorem 2.3, and we maintain that no essential assumptions have been overlooked for Theorem 2.3 to retain its validity.
>
> - The implication of Theorem 2.3 lies in whether there exists a learning algorithm capable of training or updating the model in the prescribed incremental manner (as described in Section 2). Retraining from scratch stands apart as it does not align with the principles of incremental learning and, therefore, does not qualify as a counterexample to Theorem 2.3
> - The objective of incremental learning is to devise an algorithm that achieves performance comparable to retraining from scratch, which is why we utilize it as a baseline for performance comparison in the empirical study.
> - We intentionally omitted some minor technical details from Theorem 2.3, as indicated by our note that it is an informal version. The formal version is provided in the appendix. However, we do not believe we have overlooked any crucial assumptions for Theorem 2.3 to hold.
>
> **Q4.  Why the proposed method can mitigate the structural shift? Isn’t this dependent on the data nature that we have no way to control?**
>
> We can not control structural shifts within the data but we can manage and mitigate the impact of structural shifts on the model's performance.
>
> - The key concept underpinning our proposed approach to mitigating the effects of structural shifts is our replay strategy (as detailed in Section 3.2). This strategy employs the principle of importance reweighting, which works to reduce the distance between sample representations in the latent space. This approach is specifically engineered to counteract the adverse effects of structural shifts.
>
> **Q5. Does Theorem 2.3 contradict with graph unlearning literature [1,2]?**
>
> Theorem 2.3 does not contradict the graph unlearning literature.
>
> - Graph unlearning and graph incremental learning represent two distinct problems, despite the presence of some shared terminology. Both research areas have extensive bodies of existing work. The objective of graph incremental learning is to update or train the model with new data without "forgetting" past knowledge. In contrast, graph unlearning seeks to remove specific target information from the model.
> - Both the form and the objective of the learning algorithms sought by these two problems are different, and therefore, Theorem 2.3 does not contradict the graph unlearning literature.
>
> Thank you once more for comments. We hope that our response has addressed your concerns and provided clarifications that further emphasize the significance and contributions of our work.
>
> [1] Efficient Model Updates for Approximate Unlearning of Graph-Structured Data, Chien et al. ICLR 2023.
>
> [2] Efficiently Forgetting What You Have Learned in Graph Representation Learning via Projection, Cong et al. AISTATS 2023.

---

> ### Author Response · Authors · 2023-11-19
>
> Thank you again for the feedback on our paper. We hope that our responses have addressed your inquiries and concerns. If this is not the case, please inform us and we would be glad to engage in further discussion.

---

### Official Review · Reviewer_MQqi · 2023-10-30

**Soundness:** 3 good
**Presentation:** 3 good
**Contribution:** 3 good
**Rating:** 6
**Confidence:** 5

**Summary:**

This paper presents a theoretical examination of the learnability of Node-wise Graph Incremental Learning (NGIL) in dynamical settings. Specifically, the paper presents that NGIL is not always learnable under uncontrolled structural changes.
Based on this analysis, the paper presents a technique, the Structure-Evolution-Aware Experience Replay (SEA-ER)
to control structural shifts with a sample selection that uses topological information of the GNN with importance re-weighting.
In the experiments, three real-world datasets and synthetic data are used to evaluate the proposed method (SEA-ER) and compare it to existing experience replay NGIL frameworks. It evaluates the impact of structural shift (dynamics of graph structure) and that the distortion rate is small for the datasets. Finally, the paper also presents a meta-analysis of the model with the corresponding ablation study on the size of the experience replay buffer.

**Strengths:**

This paper has several strong points. The NGIL learnability problem presents an interesting re-interpretation of the effect of cyclic probabilistic dependencies of nodes and node attributes in an evolving graph. Creating a formulation that is relatively agnostic to the mechanics of how those probabilities were produced is interesting. Theorem 2.3 is the main theoretical contribution of the paper and has some intuitive components to it, particularly concerning the fact that the impossibility of learning unconstrained dynamics.  However, this is in itself a subtle line, since the probabilistic causes for this are not fully described in the paper. Other strengths of the paper include its organization (although the clarity could be improved) and the meta-analysis.

**Weaknesses:**

The main weakness of this paper is that central to the argument is the content of Theorem 2.3. I have my reservations about this result not because it may not be true (I think is true) but because this result could be traced back to the cyclic dependencies of the node probabilities that are ultimately the contributors to the structural shift. However, this is not indicated in the paper. The paper also has several imprecisions in the descriptions for example, the metric $r_{i,j}$ is only briefly indicated to be accuracy in the "Evaluation Metric" subsection between parentheses with a "e.g." before stating it. Thus, it reads like an example and not a definitive fact. However, this is not fully confirmed later in the paper. The choice of accuracy as a metric, itself could be considered a little problematic for the problem at hand. A graph is a sparse topological mathematical entity and accuracy alone may not be the most appropriate for the task. Finally, since Theorem 2.3 is the core of the contribution the proof cannot be relegated to the appendix.

Minor comment. The caption for the Figure in page 8 is missing.

**Questions:**

I would appreciate if you could clarify the points I raised in the weakness section above.

---

> ### Author Response · Authors · 2023-11-16
>
> Thank you for the insightful feedback and comments. Please allow us to address and clarify the concerns as follows:
>
> **W1. the result can be traced back to cyclic dependencies of the node probabilities that are ultimately the contributors to the structural shift.**
>
> Our formulation and analysis do not neglect the "cyclic dependency of node probabilities" (dependency among the node distributions).
>
> - A common approach for the theoretical analysis of graph neural networks (GNNs) is to formulate the problem from the perspective of ego graphs[2,3]. From the GNN perspective, this formulation provides sufficient information (input to the GNN) for inferring a target vertex and thus forms a Markov blanket. In this formulation, the dependency among node distributions is manifested by the shared underlying graph structure.
> - In our paper, we embrace the ego-graph formulation and utilize the symmetric difference measure (defined in Section 2.2) to quantitatively capture and account for the dependency among different tasks or training sessions, which stems from the "cyclic dependency of node probability.”
>
> In summary, our formulation and analysis does not neglect the "cyclic dependency of node probabilities" but rather adopts a common approach to formulate it differently for analytical purposes.
>
> **W2. Proof of Them 2.3 in the main paper**
>
> The decision to defer the proof of Theorem 2.3 to the Appendix is motivated by two main considerations:
>
> - Space Constraint:
>     - The impossibility result is just one of the two key contributions of this paper. We need to allocate space to describe the other contribution, which is our novel experience replay method tailored to address NGIL with structural shifts.
> - Readability:
>     - Proving Theorem 2.3 requires introducing additional formalism and definitions from statistical learning theory. This extra formalism can be demanding for some readers and may divert their attention from the main message of the paper.
>     - We believe that the implications of Theorem 2.3 are more intriguing and valuable than the additional formalism we used to prove it.
>
>     In light of these considerations, we believe that deferring the formal proof to the Appendix is the most effective approach to balance readability and comprehensiveness.
>
> **W3. description and properness of evaluation metric**
>
> - Firstly, we would like to highlight that the evaluation metrics used in our empirical study are standard and widely accepted for benchmarking NGIL [1].
> - We appologize for the confusing wording and have it fixed in the revised version.
> - We agree that, given the nature of graph data, there may exist more suitable evaluation metrics. The exploration of what constitutes a superior evaluation metric and its optimal form is indeed an interesting open question. However, it is not the focus of this paper.
>
> **C1. The caption for the Figure in page 8 is missing.**
>
> - Thank you for pointing out missing caption. We have revised this in the updated paper.
>
> Thank you once more for comments. We hope that our response has addressed your concerns and provided clarifications that further emphasize the significance and contributions of our work.
>
> [1] ZHANG, Xikun, Dongjin Song, and Dacheng Tao. "CGLB: Benchmark Tasks for Continual Graph Learning." *Thirty-sixth Conference on Neural Information Processing Systems Datasets and Benchmarks Track*. 2022.
>
> [2]Ma, Jiaqi, Junwei Deng, and Qiaozhu Mei. "Subgroup generalization and fairness of graph neural networks." *Advances in Neural Information Processing Systems* 34 (2021): 1048-1061.
>
> [3] Wu, Qitian, et al. "Handling Distribution Shifts on Graphs: An Invariance Perspective." *International Conference on Learning Representations*. 2021.

---

> > ### Comment · Reviewer_MQqi · 2023-11-22
> > **Thank you for your answers.**
> >
> > I thank you the authors for taking the time to reply to my comments. Your answer to W1 clarifies my concern about the dependencies of the models. However, I still think that work needs to be done for W2 and W3. In the case of W2, while it is generally the case that the implications of a theorem are important, showing the validity of the theorem is also important. I suggest including a proof sketch at least, if the space is a constraint. With respect to W3, the clarification that you include is good. However, accuracy, FAP, FAF, are not the best performance metric because the graphs you use (CoraFull, OGB-Arxiv, Reddit) have structural characteristics that can produce outlier values and since accuracy and its variants are average-based metrics, they will be highly influenced by these outliers (for that same reason ER models may not be the best choice for these datastes). The paper you cited as a reference only listed accuracy as an option, but clearly that is not the best choice here. A better choice could be ROC-AUC matrices as suggested by the same paper you cited (Zhang et al. 2022). For that reason, I will keep my score.

---

> ### Author Response · Authors · 2023-11-19
>
> Thank you again for the feedback on our paper. We hope that our responses have addressed your inquiries and concerns. If this is not the case, please inform us and we would be glad to engage in further discussion.

---

### Official Review · Reviewer_Fnb6 · 2023-10-31

**Soundness:** 2 fair
**Presentation:** 2 fair
**Contribution:** 2 fair
**Rating:** 3
**Confidence:** 4

**Summary:**

The paper focuses on the challenges posed by Graph Incremental Learning (GIL), particularly within the context of Node-wise Graph Incremental Learning (NGIL). Traditional Graph Neural Networks (GNNs) are typically modeled for static graphs. However, many real-life networks, such as citation and financial networks, are dynamic, evolving over time. This dynamic nature results in challenges like catastrophic forgetting, where newly acquired knowledge supersedes prior learning. The paper delves deeply into the learnability of NGIL, where tasks are sequential, and the graph structure changes with each new task, giving rise to what is termed a "structural shift." Experimental results from various datasets showcase the efficacy of the proposed method.

**Strengths:**

S1. The study is well-motivated, with a comprehensive review of related work.
S2. The paper's content is easy to understand and follow.
S3. Experiments conducted on real-world datasets demonstrate the effectiveness of the methods proposed.

**Weaknesses:**

W1. The problem setting is not novel. My first concern is the novelty of the problem setting. Node-wise Graph Incremental Learning has been extensively studied by previous works

W2. The technical contributions, while sound, seem limited. The method presented integrates an experience buffer and importance re-weighting to tackle challenges such as catastrophic forgetting and structural shifts. However, using an experience buffer for stream data isn't a new concept, and many works have already explored it. The reviewer finds the technical contributions of this section somewhat limited.

W3. The theoretical innovation appears to be minimal. Once the input graph is broken down into a series of ego-graphs, the definitions, formulations, and theoretical underpinnings seem like straightforward adaptations from their IID data counterparts.

W4. The writing requires refinement:
The use of "bf" before "Node-wise Graph Incremental Learning" appears to be a formatting mistake.
The definition of distortion, as presented in Definition ??, is incomplete or missing.

W5. Lack of   time complexity analysis. it would be beneficial to compare the overall time complexity of the entire framework to that of the baselines and to provide insights into the runtime of the proposed method.

**Questions:**

See the above

---

> ### Author Response · Authors · 2023-11-16
>
> Thank you for the feedback and comments. Please allow us to address and clarify the concerns as follows:
>
> **W1 & W2. The problem setting is not novel.**
>
> The novelty of our works comes from the theoretical analysis of the learnability of NGIL and the proposed experience replay method.
>
> - We discuss extensively in the introduction and related work sections that the NIGL problem has garnered increasing attention in the community and has been studied in other related works (see the related work section in the paper).
> - Rather than diminishing the significance of our work, the increasing and existing attention to the NGIL problem should highlights the importance of our results. We provide the first impossibility result linking the learnability of NGIL and structural shift. Additionally, we introduce a novel experience replay method that outperforms other experience replay methods for NGIL under structural shift.
>
> **W2. The technical contributions, while sound, seem limited.**
>
> We respectfully disagree with the reviewer's assessment of the technical contributions of our study and would like to clarify a few points to assert that our contribution is not, in fact, limited.
>
> - First, it's important to note that our paper has two main contributions: 1) the first impossibility result for NGIL (also see the response to W1) and 2) a novel experience replay method.
> - Experience replay is a widely used framework for incremental learning, and there is a substantial body of work dedicated to exploring different experience replay strategies (as discussed in the related works and baselines of our empirical study).
> - The experience replay method proposed in the paper is novel. It is designed with the insight from the theoretical analysis and tailored for NGIL with structural shift. It demonstrates state-of-the-art performance over other experience replay methods in the NGIL problem.
>
> In conclusion, considering the heightened attention directed towards NGIL (as indicated by the reviewer), and considering the theoretical and methodological contributions of this paper to NGIL, we assert that the technical contribution of our paper is sufficient.
>
> **W3. Theoretical setting and  IID data counterparts**
>
> The theoretical setting and problem formulation do not reduce to the I.I.D data counterparts
>
> - Formulating the learning problem from the perspective of ego graphs is a common approach for the theoretical analysis of graph neural networks (e.g., [1-3]). In this formulation, the dependency among vertex is manifested by the shared underlying graph structure. Therefore, this can not be deemed as an I.I.D data setting.
> - The ``NON. I.I.D ness'' of the setting is also reflected by the analysis of the structural shift, which is the dependency of data sampling across different tasks/training sessions. Such dependency does not occur in the I.I.D data setting (e.g., incremental learning in CV) and is unique to the NGIL problem.
>
> **W4. use of ``\bf'' to highlight and definition of distortion**
>
> - use of ``\bf''
>     - The use of **`\bf`** was a deliberate choice to emphasize and draw the reader's attention to important results or concepts.
>     - This formatting convention has been widely adopted and accepted in machine-learning conference papers to enhance readability.
> - Definition of distortion
>     - The definition and description of distortion presented in Section 3.3 are complete.
>     - The definition of distortion was originally presented under the definition of environment and was later changed due to space constraints. This alteration resulted in some cross-referencing issues in the appendix. We appreciate your observation, and we have corrected this in the revised version of the paper.
>
>
> **W5. time complexity**
>
> - We have provided the complexity analysis in Appendix F. It's worth noting that all the different experience replay methods exhibit the same order of complexity, O(N), where N represents the number of vertices.
>
> Thank you once more for the comments. We hope that our response has addressed your concerns and provided clarifications that further emphasize the significance and contributions of our work.
>
> [1] Zhu, Qi, et al. "Transfer learning of graph neural networks with ego-graph information maximization." *Advances in Neural Information Processing Systems* 34 (2021): 1766-1779.
>
> [2]Ma, Jiaqi, Junwei Deng, and Qiaozhu Mei. "Subgroup generalization and fairness of graph neural networks." Advances in Neural Information Processing Systems 34 (2021): 1048-1061.
>
> [3] Wu, Qitian, et al. "Handling Distribution Shifts on Graphs: An Invariance Perspective." International Conference on Learning Representations. 2021.

---

> ### Author Response · Authors · 2023-11-19
>
> Thank you again for the feedback on our paper. We hope that our responses have addressed your inquiries and concerns. If this is not the case, please inform us and we would be glad to engage in further discussion.

---

### Official Review · Reviewer_tgji · 2023-11-01

**Soundness:** 2 fair
**Presentation:** 1 poor
**Contribution:** 2 fair
**Rating:** 5
**Confidence:** 4

**Summary:**

The paper is about incremental learning of graphs where increment happens in terms of nodes. They propose a plan to solve catastrophic forgetting in incremental learning. The idea is to subsample from historic evidence and reuse them as replay. The paper also provides theoretical analysis of the method.

**Strengths:**

1. Learning from graph in incremental setup is an important problem.
2. The idea of reusing older samples as revision seems ok.
3. The authors have provided some theoretical analysis too.

**Weaknesses:**

1. The paper is written with unnecessary formalism in some cases which makes it harder to read.
2. The main idea and the rational could have been presented in a more straightforward manner.

3. However the main concern is using some samples again and again. Although it may appear to be replay or revision but it has to be critically analysed how revisiting some samples is justified.
4. It is not clear if there is some unlearning and relearning effect is there or not.
5. It is also not clear how such replays deviate the overall learning objective.
6. How does the objective or the learning path change with respect to the order of increments of the graphs ?
7. How does the final solution change if all increments are available at the same time ?

**Questions:**

Please refer to the weakness part.

---

> ### Author Response · Authors · 2023-11-16
>
> Thank you for the feedback and comments. Please allow us to address and clarify your concerns as follows:
>
> Firstly, we'd like to briefly reiterate the incremental problem (as described in Sec.1&2), and our understanding of the unlearning problem:
>
> - Incremental learning is concerned with a setting that involves multiple training sessions or tasks. The objective is to develop a learning algorithm that can iteratively update the model with new data without "forgetting" previous knowledge. Hence, the incremental problem seeks an algorithm that takes a model parameter (already trained with some data) and new data, and outputs an updated model parameter that performs well on both new and old data.
> - Conversely, the unlearning problem focuses on removing specific information or knowledge from a trained model. The goal here is to find a learning algorithm that inputs a model parameter and (possibly) descriptors of the information to be removed, and outputs an updated model parameter that ``forget'' on the targeted information but maintains performance on the rest of the data.
>
> Although both paradigms concern the retention and removal of information, their settings, objectives, and forms of learning algorithms are distinctly different, making them orthogonal paradigms.
>
> **W1 & W2 Unnecessary formalism and straightforward presentation**
>
> We respectfully disagree that the formalism presented in the paper is unnecessary.
>
> - A key motivation and contribution of this paper is to formally formulate the NGIL problem within the statistical learning framework. This formalism is essential for two reasons:
>     - Given the inherent complexity and vague terminology associated with NGIL, formalism is crucial for precisely defining the problem and distinguishing it from similar settings, such as unlearning.
>     - The formalism, rooted in elements from statistical learning, is a necessary foundation for formally deriving the learnability of the problem.
> - We have made an effort to simplify the formalism for readability and believe its current presentation is essential.
>
> We would appreciate it if the reviewer could specify which aspects of the formalism (definition or setting) could be considered unnecessary for our results, as this would help us make targeted improvements.
>
> **W3. The main concern is using some samples again and again. Although it may appear to be replay or revision but it has to be critically analysed how revisiting some samples is justified.**
>
> - Experience replay (revisiting or replaying a small set of important samples) is one of the most effective and standard frameworks for tackling catastrophic forgetting in incremental learning [2] (also see a detailed discussion in the related work section). Instead of retraining from scratch, experience replay only needs to replay or revisit a small set of samples for consolidating knowledge and information, achieving comparable performance to retraining the model with complete data. This is a huge gain in efficiency.
> - Experience replay was originally inspired by human learning theory, which uses repeat visits to important samples to consolidate knowledge and information. The idea of experience replay has been widely adopted in incremental learning[2](also see a detailed discussion in the related work section for its application to incremental learning in CV and NLP). Many different experience methods have been developed to tailor incremental learning with different settings.
>
> **W4.  It is not clear if there is some unlearning and relearning effect is there or not.**
>
> - Catastrophic forgetting (or ``unlearning'') is a well-documented phenomenon in incremental learning studies, including NGIL contexts [1, 2]. This is evidenced in Table 1 by comparing the difference between the Bare model and Joint Training, where the Bare model shows significant performance degradation due to catastrophic forgetting.
> - Our proposed method's effectiveness (the ``relearning effect'') is demonstrated by performance retention in the NGIL setting (as defined in Eq. (1)) and is measured using standard NGIL evaluation metrics (described in Sec. 4).  The performance of our proposed method, along with other experience replay methods, is also presented in Table 1 and is reflected by the improvement over the Bare model.
>
> The above responses are based on our interpretation of what the reviewer means by "unlearning and relearning effect," as we have not used these terms anywhere in the paper. Please let us know if "unlearning and relearning effect" has a different interpretation.

---

> > ### Author Response · Authors · 2023-11-16
> >
> > **W5. It is also not clear how such replays deviate the overall learning objective.**
> >
> > - As discussed in Section 3.2, existing experience replay methods for NGIL typically employ Eq. (4) as their learning objective. However, based on insights into structural shifts, we propose a novel replay strategy (learning objective) that employs importance re-weighting and is manifested in Eq. (5).
> >
> > The above responses are based on our interpretation of what the reviewer means by the "overall learning objective." Please clarify if the "overall learning objective" has a different meaning.
> >
> > **W6. How does the objective or the learning path change with respect to the order of increments of the graphs ?**
> >
> > - The primary metric used to evaluate NGIL is the average performance of the model on all tasks after all training sessions. In theory, while the order of graph increments may affect the performance of individual tasks, it should not have a significant impact on the overall performance metric. In our empirical study, we followed the recent benchmark paper to create the NGIL setting[2].
> > - In response to this, we have included additional experiments to showcase the learning dynamics under different permutations of graph increments in Appendix D. The results align with the theoretical predictions above.
> >
> > **W7. How does the final solution change if all increments are available at the same time ?**
> > We do not believe that having all increments available at the same time would significantly impact our proposed solution or undermine the contributions of our work.
> >
> > - If all increments are available from the beginning, this setting departs from the incremental learning scenario we consider in our paper and can be reduced to a standard machine learning paradigm.
> > - If all increments become available at the second task stage, it represents a two-stage or two-task NGIL problem which corresponds exactly with the setting of our analysis and method.
> >
> > We hope our responses have effectively addressed your concerns and provided clarifications that further emphasize the significance and contributions of our work.
> >
> > [1] Matthias Delange, Rahaf Aljundi, Marc Masana, Sarah Parisot, Xu Jia, Ales Leonardis, Greg
> > Slabaugh, and Tinne Tuytelaars. A continual learning survey: Defying forgetting in classification
> > tasks. IEEE Transactions on Pattern Analysis and Machine Intelligence, 2021.
> >
> > [2] ZHANG, Xikun, Dongjin Song, and Dacheng Tao. "CGLB: Benchmark Tasks for Continual Graph Learning." *Thirty-sixth Conference on Neural Information Processing Systems Datasets and Benchmarks Track*. 2022.

---

> ### Author Response · Authors · 2023-11-19
>
> Thank you again for the feedback on our paper. We hope that our responses have addressed your inquiries and concerns. If this is not the case, please inform us and we would be glad to engage in further discussion.

---

### Author Response · Authors · 2023-11-16
**Summary of the updates in the revised paper**

In response to the reviewers' feedback, we have incorporated the following updates into the revised paper:

- Added experimental results that illustrate the effect of different order of increments
- Fixed the dangling reference caused by definition environment of distortion
- Added missing caption for the Figure in page 8

---

### Meta-Review · Area_Chair_Yzkq · 2023-12-09

**Metareview:**

This paper addresses the challenges of incremental learning in dynamic graphs, where nodes arrive sequentially and the graph structure evolves over time. The authors explore the learnability of Node-wise Graph Incremental Learning (NGIL) under uncontrolled structural changes and propose a replay-based method called Structure-Evolution-Aware Experience Replay (SEA-ER) to mitigate the issue of catastrophic forgetting. SEA-ER leverages topological information and importance re-weighting to control structural shift and improve training stability.

Strengths: The studied problem is of importance. The extensive experiments demonstrate that the proposed method outperforms prior works.

Weaknesses: My primary concern is regarding the novelty of the paper. I have briefly reviewed the proof of theorem 2.3 in the appendix. The authors resort to the tools in the theory of domain adaptation, which is standard knowledge for readers familiar with the theory of domain adaptation and transfer learning. I partially agree with the authors' response to reviewer Fnb6's W3 that the setting is different from the IID setting, but it appears to me that the theoretical underpinnings are a straightforward application of the theory of domain adaptation. Furthermore, since this is a negative result, I recommend that the authors emphasize positive results as a major contribution of the paper. The authors appear to have misunderstood reviewer vspK's Q1 and, as a result, did not provide an adequate response.

Regarding the novelty of the proposed method, I agree with reviewer Fnb6's W2: "The method presented integrates an experience buffer and importance re-weighting to address challenges such as catastrophic forgetting and structural shifts. However, using an experience buffer for stream data is not a new concept, and many works have already explored it. The reviewer finds the technical contributions of this section somewhat limited." Equation (6) is also a well-known technique (Maximum Mean Discrepancy-based distribution matching) in transfer learning. Furthermore, while appendix F.3 COMPLEXITY ANALYSIS states that the time complexity is O(number of vertices), there is no proof of this result. I do not believe that "The time complexity of our algorithm is readily apparent from its procedural steps." Moreover, what is the time complexity of solving equation (6)? In addition to the time complexity, more analysis of the running time in practice should be included. I am concerned that the authors' method may be slower than the baseline, possibly due to equation (6).

**Justification For Why Not Higher Score:**

See the weaknesses above.

**Justification For Why Not Lower Score:**

N/A

---

### Decision · Program_Chairs · 2024-01-16

Reject